# Improved Sample Complexity for Private Nonsmooth Nonconvex Optimization

**Guy Kornowski** [1]  **Daogao Liu** [2]  **Kunal Talwar** [3]

## Abstract

We study differentially private (DP) optimization algorithms for stochastic and empirical objectives which are neither smooth nor convex, and propose methods that return a Goldstein-stationary point with sample complexity bounds that improve on existing works. We start by providing a single-pass $(\varepsilon, \delta)$-DP algorithm that returns an $(\alpha, \beta)$-stationary point as long as the dataset is of size $\widetilde{\Omega}(\sqrt{d}/\alpha\beta^3 + d/\varepsilon\alpha\beta^2)$, which is $\Omega(\sqrt{d})$ times smaller than the algorithm of Zhang et al. (2024) for this task, where $d$ is the dimension. We then provide a multi-pass polynomial time algorithm which further improves the sample complexity to $\widetilde{\Omega}\left(d/\beta^2 + d^{3/4}/\varepsilon\alpha^{1/2}\beta^{3/2}\right)$, by designing a sample efficient ERM algorithm, and proving that Goldstein-stationary points generalize from the empirical loss to the population loss.

## 1. Introduction

We consider optimization problems in which the loss function is stochastic or empirical, of the form

$$F(x) := \mathop{\mathbb{E}}_{\xi \sim \mathcal{P}} [f(x; \xi)], \qquad \text{(stochastic)}$$

$$\widehat{F}^{\mathcal{D}}(x) := \frac{1}{n} \sum_{i=1}^{n} f(x; \xi_i), \qquad \text{(ERM)}$$

where $\mathcal{P}$ is the population distribution from which we sample a dataset $\mathcal{D} = (\xi_1, \ldots, \xi_n) \sim \mathcal{P}^n$, and the component functions $f(\cdot; \xi) : \mathbb{R}^d \to \mathbb{R}$ may be neither smooth nor convex. Such problems are ubiquitous throughout machine learning, where losses given by deep-learning based models give rise to highly nonsmooth nonconvex (NSNC) landscapes.

Due to its fundamental importance in modern machine learning, the field of nonconvex optimization has received substantial attention in recent years. Moving away from the classical regime of convex optimization, many works aimed at understanding the complexity of producing approximate-stationary points (with small gradient norm) for smooth nonconvex functions (Ghadimi & Lan, 2013; Fang et al., 2018; Carmon et al., 2020; Arjevani et al., 2023). However, smoothness rarely holds in modern practice, posing a major challenge for large and highly expressive models such as deep neural networks. Indeed, using ReLUs and MaxPool layers is common practice (LeCun et al., 2015), and moreover, stochastic gradient descent (SGD) with a large batch size tends to converge to sharp minima (Keskar et al., 2017)

As it turns out, without smoothness, it is actually impossible to directly minimize the gradient norm without suffering from an exponential-dimension dependent runtime in the worst case (Kornowski & Shamir, 2022). Nonetheless, a nuanced notion coined as Goldstein-stationarity (Goldstein, 1977), has been shown in recent years to enable favorable guarantees. Roughly speaking, a point $x \in \mathbb{R}^d$ is called an $(\alpha, \beta)$-Goldstein stationary point (or simply $(\alpha, \beta)$-stationary) if there exists a convex combination of gradients in the $\alpha$-ball around $x$ whose norm is at most $\beta$.[1] Following the groundbreaking work of Zhang et al. (2020), a surge of works study NSNC optimization through the lens of Goldstein stationarity, with associated finite-time guarantees (Davis et al., 2022; Lin et al., 2022; Cutkosky et al., 2023; Jordan et al., 2023; Kong & Lewis, 2023; Grimmer & Jia, 2024; Kornowski & Shamir, 2024; Tian & So, 2024).

In this work, we study NSNC optimization problems under the additional constraint of differential privacy (DP) (Dwork et al., 2006). With the ever-growing deployment of ML models in various domains, the privacy of the data on which models are trained is a major concern. Accordingly, DP optimization is an extremely well-studied problem, with a vast literature focusing on functions that are assumed to be either convex or smooth (Bassily et al., 2014; Wang et al., 2017; Bassily et al., 2019; Wang et al., 2019; Feldman et al.,

---

[1]Weizmann Institute of Science; work done while interning at Apple. [2]Google; work done while at University of Washington and Apple. [3]Apple. Correspondence to: Guy Kornowski <guy.kornowski@weizmann.ac.il>, Daogao Liu <liudaogao@gmail.com>.

*Proceedings of the 42nd International Conference on Machine Learning*, Vancouver, Canada. PMLR 267, 2025. Copyright 2025 by the author(s).

---

[1]Previous works typically use the notational convention $(\delta, \varepsilon)$-stationarity instead of $(\alpha, \beta)$, namely where $\delta$ is the radius (instead of $\alpha$) and $\varepsilon$ is the norm bound (instead of $\beta$). We depart from this notational convention in order to avoid confusion with the standard privacy notation of $(\varepsilon, \delta)$-DP.

2020; Gopi et al., 2022; Arora et al., 2023; Carmon et al., 2023; Liu et al., 2024).

The fundamental investigation in this literature is the privacy-utility trade-off, that is, assessing the minimal dataset size $n$ (referred to as the sample complexity) required in order to optimize the loss up to some error under DP. Being able to improve utility while using less samples has significant consequences, as in applications the amount of available data is a serious bottleneck, or arguably soon to become one (Villalobos et al., 2024).

For NSNC DP optimization, to the best of our knowledge the only existing result is by Zhang et al. (2024), which provided a zero-order algorithm (namely, utilizing only function value evaluations of $f(\,\cdot\,;\xi)$) that preforms a single pass over the dataset and returns an $(\alpha, \beta)$-stationary point of $F$ under $(\varepsilon, \delta)$-DP as long as

$$n = \widetilde{\Omega}\left(\frac{d}{\alpha\beta^3} + \frac{d^{3/2}}{\varepsilon\alpha\beta^2}\right). \tag{1}$$

## 1.1. Our contributions

In this paper, we provide several new algorithms for NSNC DP optimization, which improve the previously best-known sample complexity for this task. Equivalently, given the same amount of data, they provide better utility. For consistency with the previous result by Zhang et al. (2024), throughout most of this paper we propose and analyze zero-order algorithms, yet we later generalize our results to accommodate first-order (i.e. gradient) oracles.

Our contributions, summarized in Table 1, are as follows:

1. **Improved single-pass algorithm (Theorem 3.1):** We provide an $(\varepsilon, \delta)$-DP algorithm that preforms a single pass over that dataset, and returns an $(\alpha, \beta)$-stationary point as long as

$$n = \widetilde{\Omega}\left(\frac{\sqrt{d}}{\alpha\beta^3} + \frac{d}{\varepsilon\alpha\beta^2}\right), \tag{2}$$

which is $\Omega(\sqrt{d})$ times smaller than (1). Notably, the "non-private" term $\sqrt{d}/\alpha\beta^3$ has sublinear dimension dependence, as opposed to Eq. (1), which was erroneously claimed impossible by previous work (see Remark 3.2).

2. **Better multi-pass algorithm (Theorem 4.1):** In order to further improve the sample complexity, we move to consider ERM algorithms that go over the data polynomially many times, which we will later argue generalize to the population loss. To that end, we provide an $(\varepsilon, \delta)$-DP ERM algorithm, that returns an $(\alpha, \beta)$-Goldstein stationary point of $\widehat{F}^{\mathcal{D}}$ as long as

$$n = \widetilde{\Omega}\left(\frac{d^{3/4}}{\varepsilon\alpha^{1/2}\beta^{3/2}}\right). \tag{3}$$

| Sample complexity | empirical | stochastic |
|---|---|---|
| (Zhang et al., 2024) (SP) | $\frac{d}{\alpha\beta^3} + \frac{d^{3/2}}{\varepsilon\alpha\beta^2}$ | |
| Theorem 3.1 (SP) | $\frac{\sqrt{d}}{\alpha\beta^3} + \frac{d}{\varepsilon\alpha\beta^2}$ | |
| Theorem 4.1 (MP) | $\frac{d^{3/4}}{\varepsilon\alpha^{1/2}\beta^{3/2}}$ | $\frac{d}{\beta^2} + \frac{d^{3/4}}{\varepsilon\alpha^{1/2}\beta^{3/2}}$ |

*Table 1.* Main results (ignoring dependence on Lipschitz constant, initialization, and log terms). SP=Single pass, MP=Multi-pass.

Notably, Eq. (3) substantially improves Eq. (2) (and thus, Eq. (1)) in parameter regimes of interest (small $\varepsilon, \alpha, \beta$, large $d$) with respect to the dimension and accuracy parameters, and in particular is the first algorithm to perform private ERM with sublinear dimension-dependent sample complexity for NSNC objectives.

In order to utilize our empirical algorithm for stochastic objectives, we must argue that Goldstein-stationarity generalizes from the ERM to the population. No such result appears in the literature, thus we proceed to prove it:

- **Additional contribution: generalizing from ERM to population in NSNC optimization (Proposition 5.1).** We show that with high probability, any $(\alpha, \widehat{\beta})$-stationary point of $\widehat{F}^{\mathcal{D}}$ is an $(\alpha, \beta)$-stationary point of $F$, for $\beta = \widehat{\beta} + \widetilde{O}(\sqrt{d/n})$. Hence, the empirical guarantee Eq. (3) generalizes to stochastic losses with an additional $d/\beta^2$ additive term in $n$ (up to log terms).

3. **First-order algorithm with reduced oracle complexity (Theorem 6.1):** We provide a first-order (i.e., gradient-based) algorithm with the same sample complexity derived thus far, which requires $\widetilde{\Omega}(d^2)$ times less oracle calls compared to the zero-order case. Overall, this establishes the best-known algorithm for NSNC DP both in terms of sample efficiency and oracle efficiency.

## 1.2. Our techniques

One of the main techniques we use is constructing a better gradient estimator with an improved effective sensitivity. We consider the zero-order gradient estimator

$$\tilde{\nabla} f_\alpha(x; \xi) = \frac{1}{m}\sum_{j=1}^{m} \frac{d}{2\alpha}(f(x + \alpha y_j; \xi) - f(x - \alpha y_j; \xi)) \tag{4}$$

for $(y_i)_{i=1}^m \overset{iid}{\sim} \mathrm{Unif}(\mathbb{S}^{d-1})$, which is an unbiased estimator of the smoothed gradient $\nabla f_\alpha(x; \xi)$ (cf. Section 2 for details), to which we then apply variance reduction. Zhang

et al. (2024) considered this oracle specifically with $m = d$, for which it is easy to bound the estimator's sensitivity over neighboring minibatches $\xi_{1:B}, \xi'_{1:B}$ of size $B$ by

$$\left\| \frac{1}{B} \sum_{i=1}^{B} \tilde{\nabla} f_\alpha(x; \xi_i) - \frac{1}{B} \sum_{i=1}^{B} \tilde{\nabla} f_\alpha(x; \xi'_i) \right\| \le \frac{Ld}{B}. \quad (5)$$

Our key observation is that while this is indeed the worst-case sensitivity, we can get substantially lower sensitivity *with high probability*: For sufficiently large $m$, standard sub-Gaussian concentration bounds ensure that $\tilde{\nabla} f_\alpha(x; \xi_i) \approx \nabla f_\alpha(x; \xi_i)$ with high probability, and hence under this event we show the sensitivity over a mini-batch can be decreased to an order of $\frac{L}{B}$. This is a factor of $d$ smaller than Eq. (5), thus we can add significantly less noise in order to privatize, leading to faster convergence to stationarity.

## 2. Preliminaries

**Notation.** We denote by $\langle \cdot, \cdot \rangle, \|\cdot\|$ the standard Euclidean dot product and its induced norm. For $x \in \mathbb{R}^d$ and $\alpha > 0$, we denote by $\mathbb{B}(x, \alpha)$ the closed ball of radius $\alpha$ centered at $x$, and further denote $\mathbb{B}_\alpha := \mathbb{B}(0, \alpha)$. $\mathbb{S}^{d-1} \subset \mathbb{R}^d$ denotes the unit sphere. We make standard use of $O$-notation to hide absolute constants, $\widetilde{O}, \widetilde{\Omega}$ to hide poly-logarithmic factors, and also let $f \lesssim g$ denote $f = O(g)$.

**Nonsmooth optimization.** A function $h : \mathbb{R}^d \to \mathbb{R}$ is called $L$-Lipschitz if for all $x, y \in \mathbb{R}^d : |h(x) - h(y)| \le L\|x-y\|$. We call $h$ $H$-smooth, if $h$ is differentiable and $\nabla h$ is $H$-Lipschitz with respect to the Euclidean norm. For Lipschitz functions, the Clarke subgradient set (Clarke, 1990) can be defined as

$$\partial h(x) := \text{conv}\{g : g = \lim_{n \to \infty} \nabla h(x_n), x_n \to x\},$$

namely the convex hull of all limit points of $\nabla h(x_n)$ over sequences of differentiable points (which are a full Lebesgue-measure set by Rademacher's theorem), converging to $x$. For $\alpha \ge 0$, the Goldstein $\alpha$-subdifferential (Goldstein, 1977) is further defined as

$$\partial_\alpha h(x) := \text{conv}(\cup_{y \in \mathbb{B}(x,\alpha)} \partial h(y)),$$

and we denote the minimum-norm element of the Goldstein $\alpha$-subdifferential by

$$\overline{\partial}_\alpha h(x) := \arg\min_{g \in \partial_\alpha h(x)} \|g\|.$$

**Definition 2.1.** A point $x \in \mathbb{R}^d$ is called an $(\alpha, \beta)$-Goldstein stationary point of $h$ if $\left\| \overline{\partial}_\alpha h(x) \right\| \le \beta$.

Throughout the paper we impose the following standard Lipschitz assumption:

**Assumption 2.2.** For any $\xi$, $f(\cdot; \xi) : \mathbb{R}^d \to \mathbb{R}$ is $L$-Lipschitz (hence, so is $F$).

**Randomized smoothing.** Given any function $h : \mathbb{R}^d \to \mathbb{R}$, we denote its randomized smoothing $h_\alpha(x) := \mathbb{E}_{y \sim \mathbb{B}_\alpha} h(x+y)$. We recall the following standard properties of randomized smoothing (Flaxman et al., 2005; Yousefian et al., 2012; Duchi et al., 2012; Shamir, 2017).

**Fact 2.3** (Randomized smoothing). *Suppose $h : \mathbb{R}^d \to \mathbb{R}$ is $L$-Lipschitz. Then: (i) $h_\alpha$ is $L$-Lipschitz; (ii) $|h_\alpha(x) - h(x)| \le L\alpha$ for any $x \in \mathbb{R}^d$; (iii) $h_\alpha$ is $O(L\sqrt{d}/\alpha)$-smooth; (iv) $\nabla h_\alpha(x) = \mathbb{E}_{y \sim \mathbb{B}_\alpha}[\nabla h(x+y)] = \mathbb{E}_{y \sim \mathbb{S}^{d-1}}[\frac{d}{2\alpha}(h(x + \alpha y) - h(x - \alpha y))y]$.*

The following result shows that in order to find a Goldstein-stationary point of a function, it suffices to find a Goldstein-stationary point of its randomized smoothing:

**Lemma 2.4** (Kornowski & Shamir, 2024, Lemma 4). *Any $(\alpha, \beta)$-stationary point of $h_\alpha$ is a $(2\alpha, \beta)$-stationary point of $h$.*

**Differential privacy.** Two datasets $\mathcal{D}, \mathcal{D}' \in \text{supp}(\mathcal{P})^n$ are said to be neighboring if they differ in only one data point. A randomized algorithm $\mathcal{A} : \mathcal{Z}^n \to \mathcal{R}$ is called $(\varepsilon, \delta)$ differentially private (or $(\varepsilon, \delta)$-DP) for $\varepsilon, \delta > 0$ if for any two neighboring datasets $\mathcal{D}, \mathcal{D}'$ and measurable $E \subseteq \mathcal{R}$ in the algorithm's range, it holds that $\Pr[\mathcal{A}(\mathcal{D}) \in E] \le e^\varepsilon \Pr[\mathcal{A}(\mathcal{D}') \in E] + \delta$ (Dwork et al., 2006).

**Tree mechanism.** Next, we revisit the well-known tree mechanism, detailed in Algorithm 1. In essence, the tree mechanism enables the privatization of cumulative sums $\sum_{j=1}^{t} g_j$ for all $t \in [\Sigma]$, which, in our context, correspond to summed gradient estimators. A naive approach would add independent Gaussian noise $\zeta_j$ to each $g_j$, leading to an error that scales as $\sqrt{\Sigma}$. In contrast, the tree mechanism reduces this error to $O(\log \Sigma)$ by introducing correlated Gaussian noise. Broadly speaking, the mechanism generates a set of independent Gaussian noise values and organizes them in a tree-like structure, where each node corresponds to a specific noise value. To compute any cumulative sum $\sum_{j=1}^{t} g_j$, the mechanism selects at most $O(\log \Sigma)$ nodes (Function NODE in Algorithm 1) from the tree and privatizes the sum using the aggregated noise from these nodes (i.e., final noise TREE$(t)$ generated by Algorithm 1). The formal guarantee associated with this mechanism is provided below.

**Proposition 2.5** (Tree Mechanism Dwork et al., 2010; Chan et al., 2011; Zhang et al., 2024). *Let $\mathcal{Z}_1, \cdots, \mathcal{Z}_\Sigma$ be dataset spaces, suppose $\mathcal{X} \subseteq \mathbb{R}^d$, and let $\mathcal{M}_i : \mathcal{X}^{i-1} \times \mathcal{Z}_i \to \mathcal{X}$ be a sequence of algorithms for $i \in [\Sigma]$. Let $\text{ALG} : \mathcal{Z}^{(1:\Sigma)} \to \mathcal{X}^\Sigma$ be the algorithm that given a dataset $Z_{1:\Sigma} \in \mathcal{Z}^{(1:\Sigma)}$, sequentially computes $X_i = \sum_{j=1}^{i} \mathcal{M}_j(X_{1:j-1}, Z_j) + \text{TREE}(i)$ for $i \in [\Sigma]$, and then outputs $X_{1:\Sigma}$. Suppose for all $i \in [\Sigma]$, and neighboring $Z_{1:\Sigma}, Z'_{1:\Sigma} \in \mathcal{Z}^{(1:\Sigma)}, \|\mathcal{M}_i(X_{1:i-1}, Z_i) - \mathcal{M}_i(X_{1:i-1}, Z'_i)\| \le s$ for all auxiliary inputs $X_{1:i-1} \in \mathcal{X}^{i-1}$. Then setting $\sigma =$*

$4s\sqrt{\log \Sigma \log(1/\delta)}/\varepsilon$, *Algorithm 1 is $(\varepsilon, \delta)$-DP. Furthermore, for any $t \in [\Sigma]$ : $\mathbb{E}[\text{TREE}(t)] = 0$ and $\mathbb{E}\|\text{TREE}(t)\|^2 \lesssim d \log(\Sigma)\sigma^2$.*

In our case, the "dataset spaces" $\mathcal{Z}_i$ will be collections of possible minibatches of some determined size, and $\mathcal{M}_i$ will be a gradient estimator with respect to the sampled minibatch at some current iterate.

---

**Algorithm 1** Tree Mechanism

1: **Input:** Noise parameter $\sigma$, sequence length $\Sigma$
2: Define $\mathcal{T} := \{(u,v) : u = j \cdot 2^{\ell-1} + 1, v = (j+1) \cdot 2^{\ell-1}, 1 \le \ell \le \log \Sigma, 0 \le j \le \Sigma/2^{\ell-1} - 1\}$
3: Sample and store $\zeta_{(u,v)} \sim \mathcal{N}(0, \sigma^2)$ for all $(u,v) \in \mathcal{T}$
4: **for** $t = 1, \cdots, \Sigma$ **do**
5:    Let $\text{TREE}(t) \leftarrow \sum_{(u,v)\in\text{NODE}(t)} \zeta_{(u,v)}$
6: **end for**
7: **Return:** $\text{TREE}(t)$ for each $t \in [\Sigma]$
8:
9: **Function NODE:**
10: **Input:** index $t \in [\Sigma]$
11: Initialize $S = \{\}$ and $k = 0$
12: **for** $i = 1, \cdots, \lceil \log \Sigma \rceil$ while $k < t$ **do**
13:    Set $k' = k + 2^{\lceil \log \Sigma \rceil - i}$
14:    **if** $k' \le t$ **then**
15:       $S \leftarrow S \cup \{(k+1, k')\}, k \leftarrow k'$
16:    **end if**
17: **end for**

---

### 2.1. Base algorithm: O2NC

Similar to Zhang et al. (2024), our general algorithm is based on the so-called "Online-to-Non-Convex conversion" (O2NC) of Cutkosky et al. (2023), which is generally an optimal method for finding Goldstein-stationary points (without the privacy constraint). We slightly modify previous proofs by disentangling the role of the variance of the gradient estimator vs. its second order moment, as follows:

**Proposition 2.6** (O2NC). *Suppose that $\mathcal{O}(\cdot)$ is a stochastic gradient oracle of some differentiable function $h : \mathbb{R}^d \to \mathbb{R}$, so that for all $z \in \mathbb{R}^d$ : $\mathbb{E}\|\mathcal{O}(z) - \nabla h(z)\|^2 \le G_0^2$ and $\mathbb{E}\|\mathcal{O}(z)\|^2 \le G_1^2$. Then running Algorithm 2 with $\eta = \frac{D}{G_1\sqrt{M}}$, $MD \le \alpha$, uses $T$ calls to $\mathcal{O}(\cdot)$, and satisfies*

$$\mathbb{E}\left\|\overline{\partial}_\alpha h(x^{\text{out}})\right\| \le \frac{h(x_0) - \inf h}{DT} + \frac{3G_1}{2\sqrt{M}} + G_0.$$

We provide a proof of Proposition 2.6 in Appendix B. Recalling that by Lemma 2.4 any $(\alpha, \beta)$-stationary point of $F_\alpha$ is a $(2\alpha, \beta)$-stationary point of $F$, we see that it is enough to design a private stochastic gradient oracle $\mathcal{O}$ of $\nabla F_\alpha$, while controlling its variance $G_0$ and second moment $G_1$. In the next sections, we show how to construct such private oracles

and derive the corresponding guarantees through Proposition 2.6. As previously remarked, throughout most of the paper, our oracles will be based on zero-order queries of the component functions $f(\cdot, \xi)$, yet in Section 6 we construct oracles with the same sample complexity using first-order queries, leading to a lower oracle complexity overall.

---

**Algorithm 2** Nonsmooth Nonconvex Algorithm (based on O2NC (Cutkosky et al., 2023))

1: **Input:** Oracle $\mathcal{O} : \mathbb{R}^d \to \mathbb{R}^d$, initialization $x_0 \in \mathbb{R}^d$, clipping parameter $D > 0$, step size $\eta > 0$, averaging length $M \in \mathbb{N}$, iteration budget $T \in \mathbb{N}$.
2: **Initialize:** $\Delta_1 = \mathbf{0}$
3: **for** $t = 1, \ldots, T$ **do**
4:    Sample $s_t \sim \text{Unif}[0,1]$
5:    $x_t = x_{t-1} + \Delta_t$
6:    $z_t = x_{t-1} + s_t \Delta_t$
7:    $\tilde{g}_t = \mathcal{O}(z_t)$
8:    $\Delta_{t+1} = \min\{1, \frac{D}{\|\Delta_t - \eta\tilde{g}_t\|}\} \cdot (\Delta_t - \eta\tilde{g}_t)$
9: **end for**
10: $K = \lfloor \frac{T}{M} \rfloor$
11: **for** $k = 1, \ldots, K$ **do**
12:    $\overline{x}_k = \frac{1}{M} \sum_{m=1}^{M} z_{(k-1)M+m}$
13: **end for**
14: Sample $x^{\text{out}} \sim \text{Unif}\{\overline{x}_1, \ldots, \overline{x}_K\}$
15: **Output:** $x^{\text{out}}$.

---

## 3. Single-pass algorithm

In this section, we consider Algorithm 3, which provides an oracle to be used in Algorithm 2. Algorithm 3 is such that throughout $T$ calls, it uses each data point once, and hence, privacy is maintained with no need for composition. The main theorem in this section is the following:

**Theorem 3.1** (Single-pass algorithm). *Suppose $F(x_0) - \inf_x F(x) \le \Phi$, that Assumption 2.2 holds, and let $\alpha, \beta, \delta, \varepsilon > 0$ such that $\alpha \le \frac{\Phi}{L}$. Then setting $B_1 = \Sigma$, $B_2 = 1$, $M = \alpha/4D$, $m = \widetilde{O}(d^2B_1^2 + \frac{d\alpha^2 B_2^2}{D^2})$, $\sigma = \widetilde{O}(\frac{L}{B_1\varepsilon} + \frac{LD\sqrt{d}}{\alpha B_2 \varepsilon})$, $\Sigma = \widetilde{\Theta}((\frac{\alpha}{\varepsilon D})^{2/3} + \frac{\alpha}{D d^{1/2}})$, $D = \widetilde{\Theta}(\min\{(\frac{\Phi^2 \alpha}{L^2 T^2})^{1/3}, (\frac{\Phi\alpha\varepsilon}{dLT})^{1/2}, (\frac{\Phi^3 \alpha^2 \varepsilon}{d^{3/2} L^3 T^3})^{1/5}, (\frac{\Phi^2 \alpha}{L^2 T^2 \sqrt{d}})^{1/3}\})$, $T = \Theta(n)$, and running Algorithm 2 with Algorithm 3 as the oracle subroutine, is $(\varepsilon, \delta)$-DP. Furthermore, its output satisfies $\mathbb{E}\|\overline{\partial}_{2\alpha} F(x^{\text{out}})\| \le \beta$ as long as*

$$n = \widetilde{\Omega}\left(\frac{\Phi L^2 \sqrt{d}}{\alpha \beta^3} + \frac{\Phi L d}{\varepsilon \alpha \beta^2}\right).$$

*Remark* 3.2. It is interesting to note that the "non-private" term $\Phi L^2 \sqrt{d}/\alpha\beta^3$ in Theorem 3.1 has sublinear dependence on the dimension $d$. Not only is this the first such result, this was even (erroneously) claimed impossible by Zhang et al. (2024). The reason for this confusion is that

**Algorithm 3** Single-pass instantiation of $\mathcal{O}(z_t)$ in Line 7 of Algorithm 2

1: **Input:** Current iterate $z_t$, time $t \in \mathbb{N}$, period length $\Sigma \in \mathbb{N}$, accuracy parameter $\alpha > 0$, batch sizes $B_1, B_2 \in \mathbb{N}$, gradient validation size $m \in \mathbb{N}$, noise level $\sigma > 0$.
2: **if** $t \bmod \Sigma = 1$ **then**
3:    Sample minibatch $S_t$ of size $B_1$ from unused samples
4:    **for** each sample $\xi_i \in S_t$ **do**
5:       Sample $y_1, \ldots, y_m \overset{iid}{\sim} \mathrm{Unif}(\mathbb{S}^{d-1})$
6:       $\tilde{\nabla} f(z_t; \xi_i) = \frac{1}{m} \sum_{j \in [m]} \frac{d}{2\alpha} (f(z_t + \alpha y_j; \xi_i) - f(z_t - \alpha y_j; \xi_i)) y_j$
7:    **end for**
8:    $g_t = \frac{1}{B_1} \sum_{\xi_i \in S_t} \tilde{\nabla} f(z_t; \xi_i)$
9: **else**
10:   Sample minibatch $S_t$ of size $B_2$ from unused samples
11:   **for** each sample $\xi_i \in S_t$ **do**
12:      Sample $y_1, \ldots, y_{2m} \overset{iid}{\sim} \mathrm{Unif}(\mathbb{B}_\alpha)$
13:      $\tilde{\nabla} f(z_t; \xi_i) = \frac{1}{m} \sum_{j \in [m]} \frac{d}{2\alpha} (f(z_t + \alpha y_j; \xi_i) - f(z_t - \alpha y_j; \xi_i)) y_j$
14:      $\tilde{\nabla} f(z_{t-1}; \xi_i) = \frac{1}{m} \sum_{j=m+1}^{2m} \frac{d}{2\alpha} (f(z_{t-1} + \alpha y_j; \xi_i) - f(z_{t-1} - \alpha y_j; \xi_i)) y_j$
15:   **end for**
16:   $g_t = g_{t-1} + \frac{1}{B_2} \sum_{\xi_i \in S_t} (\tilde{\nabla} f(z_t; \xi_i) - \tilde{\nabla} f(z_{t-1}; \xi_i))$
17: **end if**
18: **Return** $\tilde{g}_t = g_t + \mathrm{TREE}(\sigma, \Sigma)(t \bmod \Sigma)$

---

while the optimal zero-order *oracle* complexity is $d/\alpha\beta^3$ (Kornowski & Shamir, 2024), and in particular must scale linearly with the dimension (Duchi et al., 2015), the *sample* complexity might not.

*Remark* 3.3. Since Algorithm 2 uses $T$ calls to $\mathcal{O}(\cdot)$, and it easy to see that the amortized oracle complexity of $\mathcal{O}(\cdot)$ is $O(m)$, the overall oracle complexity we get is $O(Tm)$. As previously mentioned, we set $m$ to reduce the sensitivity of $\mathcal{O}(\cdot)$, leading to an improvement in sample complexity. More generally, our analysis allows trading-off sample and oracle complexities in a Pareto-front fashion. We further use this observation in Section 6, where we show that first-order oracles allow setting a substantially smaller $m$ while maintaining the same reduced sensitivity.

In the rest of the section, we will present the basic properties of this oracle in terms of sensitivity (implying the privacy), variance and second moment. We will then plug these into Algorithm 2, which enables proving Theorem 3.1. Corresponding proofs are deferred to Appendix A.

**Lemma 3.4** (Sensitivity). *Consider the gradient oracle $\mathcal{O}(\cdot)$ in Algorithm 3 when acting on two neighboring minibatches $S_t$ and $S_t'$, and correspondingly producing $g_t$ and $g_t'$, respectively. If $t \bmod \Sigma = 1$, then it holds with probability at*

least $1 - \delta/2$ that

$$\|g_t - g_t'\| \lesssim \frac{L}{B_1} + \frac{Ld\sqrt{\log(dB_1/\delta)}}{\sqrt{m}}.$$

*Otherwise, conditioned on $g_{t-1} = g_{t-1}'$, we have with probability at least $1 - \delta/2$:*

$$\|g_t - g_t'\| \lesssim \frac{L\sqrt{d}D}{\alpha B_2} + \frac{Ld\sqrt{\log(dB_1/\delta)}}{\sqrt{m}}.$$

With the sensitivity bound given by Lemma 3.4, we easily derive the privacy guarantee of our oracle from the Tree Mechanism (Proposition 2.5).

**Lemma 3.5** (Privacy). *Running Algorithm 3 with $m = O\left(\log(dB_2/\delta)(d^2 B_1^2 + \frac{d\alpha^2 B_2^2}{D^2})\right)$ and $\sigma = O\left(\frac{L\sqrt{\log(1/\delta)}}{B_1 \varepsilon} + \frac{LD\sqrt{d\log(1/\delta)}}{\alpha B_2 \varepsilon}\right)$ is $(\varepsilon, \delta)$-DP.*

We next analyze the variance and second moment of the gradient oracle.

**Lemma 3.6** (Variance). *In Algorithm 3, for all $t \in [T]$ it holds that*

$$\mathbb{E}\|\tilde{g}_t - \nabla F_\alpha(z_t)\|^2 \lesssim \frac{L^2}{B_1} + \frac{L^2 d^2}{B_1 m} + \frac{L^2 d D^2 \Sigma}{\alpha^2 B_2}$$
$$+ \sigma^2 d \log \Sigma + \frac{L^2 d^2 \Sigma}{m B_2},$$

$$\mathbb{E}\|\tilde{g}_t\|^2 \lesssim L^2 + \frac{L^2 d^2}{B_1 m} + \frac{L^2 d D^2 \Sigma}{\alpha^2 B_2}$$
$$+ \sigma^2 d \log \Sigma + \frac{L^2 d^2 \Sigma}{m B_2}.$$

Combining the ingredients that we have set up, we can derive Theorem 3.1.

*Proof of Theorem 3.1.* The privacy guarantee follows directly from Lemma 3.5, by noting that our parameter assignment implies $B_1 T/\Sigma + B_2 T = O(n)$, which allows letting $T = \Theta(n)$ while never re-using samples (hence no privacy composition is required). Therefore, it remains to show the utility bound. By applying Lemma 2.4 and Proposition 2.6, we get that

$$\mathbb{E}\|\overline{\partial}_{2\alpha} F(x^{\mathrm{out}})\| \leq \mathbb{E}\|\overline{\partial}_\alpha F_\alpha(x^{\mathrm{out}})\|$$
$$\leq \frac{F_\alpha(x_0) - \inf F_\alpha}{DT} + \frac{3G_1}{2\sqrt{M}} + G_0$$
$$\leq \frac{2\Phi}{DT} + \frac{3G_1}{2\sqrt{M}} + G_0, \tag{6}$$

where the last inequality used the fact that Assumption 2.2 and Fact 2.3 together imply that $F_\alpha(x_0) - \inf F_\alpha \leq$

$F(x_0) - \inf F + L\alpha \leq \Phi + L\alpha \leq 2\Phi$. Under our parameter assignment, Lemma 3.6 yields

$$G_1 \lesssim G_0 + L, \qquad (7)$$

which plugged into Eq. (6) gives

$$\mathbb{E}\|\overline{\partial}_{2\alpha}F(x^{\text{out}})\| = O\left(\frac{\Phi}{DT} + \frac{L}{\sqrt{M}} + G_0\right). \qquad (8)$$

Moreover, under our parameter assignment, Lemma 3.6 also gives the bound

$$G_0 \lesssim \frac{L}{\sqrt{B_1}} + \frac{LD\sqrt{d\Sigma}}{\alpha\sqrt{B_2}} + \sigma\sqrt{d\log\Sigma} \qquad (9)$$
$$= \widetilde{O}\left(\frac{L}{\sqrt{\Sigma}} + \frac{LDd^{1/2}\Sigma^{1/2}}{\alpha} + \frac{Ld^{1/2}}{\Sigma\varepsilon} + \frac{LDd}{\alpha\varepsilon}\right),$$

which propagated into Eq. (8) and recalling that $M = \Theta(\alpha/D)$ shows that

$$\mathbb{E}\|\overline{\partial}_{2\alpha}F(x^{\text{out}})\|$$
$$= \widetilde{O}\bigg(\frac{\Phi}{DT} + \frac{LD^{1/2}}{\alpha^{1/2}} + \frac{LDd^{1/2}\Sigma^{1/2}}{\alpha}$$
$$+ \frac{L}{\sqrt{\Sigma}} + \frac{Ld^{1/2}}{\Sigma\varepsilon} + \frac{LDd}{\alpha\varepsilon}\bigg).$$

Plugging our assignments of $\Sigma$ and $D$, and recalling that $n = \Theta(T)$, a straightforward calculation simplifies the bound above to

$$\mathbb{E}\|\overline{\partial}_{2\alpha}F(x^{\text{out}})\|$$
$$= \widetilde{O}\bigg(\Big(\frac{\Phi L^2\sqrt{d}}{n\alpha}\Big)^{1/3} + \Big(\frac{\Phi dL}{n\alpha\varepsilon}\Big)^{1/2} + \Big(\frac{\Phi^2 L^3 d^{3/2}}{n^2\alpha^2\varepsilon}\Big)^{1/5}\bigg).$$

Bounding by $\beta$ and solving for $n$ results in

$$n = \widetilde{\Omega}\left(\frac{\Phi L^2\sqrt{d}}{\alpha\beta^3} + \frac{\Phi Ld}{\varepsilon\alpha\beta^2} + \frac{\Phi L^{3/2}d^{3/4}}{\varepsilon^{1/2}\alpha\beta^{5/2}}\right).$$

To complete the proof, we simply note that $\frac{\Phi L^{3/2}d^{3/4}}{\varepsilon^{1/2}\alpha\beta^{5/2}} \leq \frac{\Phi L^2\sqrt{d}}{\alpha\beta^3} + \frac{\Phi Ld}{\varepsilon\alpha\beta^2}$ by the AM-GM inequality, and so the third term above is negligible.

□

## 4. Multi-pass algorithm

In this section, we consider a different oracle construction given by Algorithm 4, to be used in Algorithm 2. The main difference from the previous section is that this oracle reuses data points a polynomial number of times, and therefore cannot *directly* guarantee generalization to the stochastic objective. Instead, in this section we analyze

---

**Algorithm 4** Multi-pass instantiation of $\mathcal{O}(z_t)$ in Line 7 of Algorithm 2

1: **Input:** Current iterate $z_t$, time $t \in \mathbb{N}$, period length $\Sigma \in \mathbb{N}$, accuracy parameter $\alpha > 0$, gradient validation size $m \in \mathbb{N}$, noise levels $\sigma_1, \sigma_2 > 0$.
2: **if** $t \bmod \Sigma = 1$ **then**
3:   **for** each sample $\xi_i \in \mathcal{D}$ **do**
4:     Sample $y_1, \ldots, y_m \overset{iid}{\sim} \text{Unif}(\mathbb{S}^{d-1})$
5:     $\tilde{\nabla}f(z_t; \xi_i) = \frac{1}{m}\sum_{j\in[m]}\frac{d}{2\alpha}(f(z_t + \alpha y_j; \xi_i) - f(z_t - \alpha y_j; \xi_i))y_j$
6:   **end for**
7:   $g_t = \frac{1}{n}\sum_{\xi_i\in\mathcal{D}}\tilde{\nabla}f(z_t; \xi_i)$
8:   **Return:** $\tilde{g}_t = g_t + \chi_t$, where $\chi_t \sim \mathcal{N}(0, \sigma_1^2 I_d)$
9: **else**
10:   **for** each sample $\xi_i \in \mathcal{D}$ **do**
11:     Sample $y_1, \ldots, y_{2m} \overset{iid}{\sim} \text{Unif}(\mathbb{B}_\alpha)$
12:     $\tilde{\nabla}f(z_t; \xi_i) = \frac{1}{m}\sum_{j=1}^m\frac{d}{2\alpha}(f(z_t + \alpha y_j; \xi_i) - f(z_t - \alpha y_j; \xi_i))y_j$
13:     $\tilde{\nabla}f(z_{t-1}; \xi_i) = \frac{1}{m}\sum_{j=m+1}^{2m}\frac{d}{2\alpha}(f(z_{t-1} + \alpha y_j; \xi_i) - f(z_{t-1} - \alpha y_j; \xi_i))y_j$
14:   **end for**
15:   $g_t = \tilde{g}_{t-1} + \frac{1}{n}\sum_{\xi_i\in\mathcal{D}}(\tilde{\nabla}f(z_t; \xi_i) - \tilde{\nabla}f(z_{t-1}; \xi_i))$
16:   **Return:** $\tilde{g}_t = g_t + \chi_t$, where $\chi_t \sim \mathcal{N}(0, \sigma_2^2 I_d)$.
17: **end if**

---

the empirical objective $\widehat{F}^{\mathcal{D}}(x) := \frac{1}{n}\sum_{i=1}^n f(x; \xi_i)$. After establishing ERM results, in Section 5 we show that any empirical Goldstein-stationarity guarantee generalizes to the population loss.

Similarly to the single-pass oracle (Algorithm 3), we use randomized smoothing and variance reduction. A difference in the oracle construction is that we replace the tree mechanism with the Gaussian mechanism and apply advanced composition for the privacy analysis (since now samples are reused). The main theorem for this section is the following:

**Theorem 4.1** (Multi-pass ERM). *Suppose* $\widehat{F}^{\mathcal{D}}(x_0) - \inf_x \widehat{F}^{\mathcal{D}}(x) \leq \Phi$, *Assumption 2.2 holds, and let* $\alpha, \beta, \delta, \varepsilon > 0$ *such that* $\alpha \leq \frac{\Phi}{L}$. *Then setting* $m = \frac{L^2 d\Sigma}{n\sigma_1^2} + \frac{L^2 d}{n\sigma_2^2}$, $\sigma_1 = O(\frac{L\sqrt{T\log(1/\delta)/\Sigma}}{n\varepsilon})$, $\sigma_2 = O(\frac{LD\sqrt{Td\log(1/\delta)}}{\alpha n\varepsilon})$, $\Sigma = \widetilde{\Theta}(\frac{\alpha}{D\sqrt{d}})$, $D = \widetilde{\Theta}(\frac{\alpha^2\beta^2}{L^2})$, $T = \widetilde{\Theta}(\frac{\Phi L^2}{\alpha^2\beta^3})$, *and running Algorithm 2 with Algorithm 4 as the oracle subroutine is* $(\varepsilon, \delta)$-*DP. Furthermore, its output satisfies* $\mathbb{E}\|\overline{\partial}_{2\alpha}\widehat{F}^{\mathcal{D}}(x^{\text{out}})\| \leq \beta$ *as long as*

$$n = \widetilde{\Omega}\left(\frac{\sqrt{\Phi}Ld^{3/4}}{\varepsilon\alpha^{1/2}\beta^{3/2}}\right).$$

*Remark* 4.2. As we will show in Section 5, Theorem 4.1 also provides the same population guarantee for $\|\overline{\partial}_{2\alpha}F(x^{\text{out}})\|$ with an additional $L^2 d/\beta^2$ term (up to log factors) to the sample complexity.

To prove Theorem 4.1, we analyze the properties of the oracle given by Algorithm 4. The sensitivity of $g_t$ in Algorithm 4 directly follows from Lemma 3.4.[2] By the standard composition results of the Gaussian mechanism (e.g., Abadi et al. 2016; Mironov 2017; Kulkarni et al. 2021), we have the following privacy guarantee:

**Lemma 4.3** (Privacy). *Calling Algorithm 4 $T$ times with* $m = \frac{L^2 d\Sigma}{n\sigma_1^2} + \frac{L^2 d}{n\sigma_2^2}$, $\sigma_1 = O(\frac{L\sqrt{T\log(1/\delta)/\Sigma}}{n\varepsilon})$ *and* $\sigma_2 = O(\frac{LD\sqrt{Td\log(1/\delta)}}{\alpha n\varepsilon})$ *is* $(\varepsilon, \delta)$-*DP.*

In terms of the oracle's variance, we show:

**Lemma 4.4** (Variance). *In Algorithm 4, for any $t \in [T]$, we have*

$$\mathbb{E}\|\tilde{g}_t - \nabla F_\alpha^{\mathcal{D}}(z_t)\|^2 \lesssim \frac{L^2 d^2 \Sigma}{mn} + \sigma_1^2 d + \sigma_2^2 d\Sigma,$$

$$\mathbb{E}\|\tilde{g}_t\|^2 \lesssim L^2 + \frac{L^2 d^2 \Sigma}{mn} + \sigma_1^2 d + \sigma_2^2 d\Sigma.$$

The proof of Theorem 4.1, which we defer to Appendix A, is a combination of the two lemmas and Proposition 2.6.

## 5. Empirical to population Goldstein-stationarity

In this section, we provide a generalization result, showing that our ERM algorithm from the previous section also guarantees Goldstein-stationarity in terms of the population loss. We prove the following more general statement:

**Proposition 5.1.** *Under Assumption 2.2, suppose $\mathcal{D} \sim \mathcal{P}^n$, and consider running an algorithm on $\widehat{F}^{\mathcal{D}}$ whose (possibly randomized) output $x^{\text{out}} \in \mathcal{X} \subset \mathbb{R}^d$ is supported over a set $\mathcal{X}$ of diameter $\leq R$. Then with probability at least $1 - \zeta$ : $\|\overline{\partial}_\alpha F(x^{\text{out}})\| \leq \|\overline{\partial}_\alpha \widehat{F}^{\mathcal{D}}(x^{\text{out}})\| + \widetilde{O}\left(L\sqrt{d\log(R/\zeta)/n}\right)$.*

We remark that in all algorithms of interest, the output is known to lie in some predefined set, such as a sufficiently large ball around the initialization. As long as the diameter $R$ is polynomial in the problem parameters, the $\log(R)$ in the result above is therefore negligible. For instance, Algorithm 2 is easily verified to output a point $x^{\text{out}} \in \mathbb{B}(x_0, DT)$ (since $\|x_{t+1} - x_t\| \leq D$ for all $t$). Hence, in our use case, Proposition 5.1 ensures $\|\overline{\partial}_{2\alpha} F(x^{\text{out}})\| \leq \|\overline{\partial}_{2\alpha}\widehat{F}^{\mathcal{D}}(x^{\text{out}})\| + \beta$ for $n = \widetilde{O}(d/\beta^2)$.

## 6. Improved efficiency with gradients

In this section, our goal is to show that the zero-order algorithms presented thus far can be replaced by first-order

algorithms with the same sample complexity, and improved oracle complexity.

The idea is to replace the zero-order gradient estimator from Eq. (4) by the smoothed first-order estimator

$$\tilde{\nabla} f_\alpha(x; \xi) = \frac{1}{m} \sum_{j=1}^{m} \nabla f(x + \alpha y_j; \xi) \qquad (10)$$

for $(y_i)_{i=1}^m \overset{iid}{\sim} \text{Unif}(\mathbb{S}^{d-1})$. While this estimator has the same expectation as the zero-order variant, the key difference lies in the fact that its sub-Gaussian norm is substantially smaller, and in particular, it does not depend on $d$. Hence, smaller $m$ suffices for similar concentration. This observation enables reducing the oracle complexity, while ensuring the same sample complexity guarantee as earlier.

We fully analyze here a single-pass first-order oracle presented in Algorithm 5, which can be used in Algorithm 2, similarly to Section 3. We note that the same analysis can be applied to the multi-pass oracle of Section 4, once again by replacing Eq. (4) by Eq. (10), which we omit here for brevity. The main result in this section is the following:

**Theorem 6.1** (First-order). *Suppose $F(x_0) - \inf_x F(x) \leq \Phi$, that Assumption 2.2 holds, and let $\alpha, \beta, \delta, \varepsilon > 0$ such that $\alpha \leq \frac{\Phi}{L}$. Then setting $B_1 = \Sigma$, $B_2 = 1$, $M = \alpha/4D$, $m = \widetilde{O}(\frac{B_2^2 \alpha^2}{D^2 d})$, $\sigma = \widetilde{O}(\frac{L}{B_1 \varepsilon} + \frac{LD\sqrt{d}}{\alpha B_2 \varepsilon})$, $\Sigma = \widetilde{\Theta}((\frac{\alpha}{\varepsilon D})^{2/3} + \frac{\alpha}{Dd^{1/2}})$, $D = \widetilde{\Theta}(\min\{(\frac{\Phi^2 \alpha}{L^2 T^2})^{1/3}, (\frac{\Phi \alpha \varepsilon}{dLT})^{1/2}, (\frac{\Phi^3 \alpha^2 \varepsilon}{d^{3/2} L^3 T^3})^{1/5}, (\frac{\Phi^2 \alpha}{L^2 T^2 \sqrt{d}})^{1/3}\})$, $T = \Theta(n)$, and running Algorithm 2 with Algorithm 5 as the oracle subroutine, is $(\varepsilon, \delta)$-DP. Furthermore, its output satisfies $\mathbb{E}\|\overline{\partial}_{2\alpha} F(x^{\text{out}})\| \leq \beta$ as long as*

$$n = \widetilde{\Omega}\left(\frac{\Phi L^2 \sqrt{d}}{\alpha\beta^3} + \frac{\Phi Ld}{\varepsilon\alpha\beta^2}\right).$$

*Remark* 6.2. Compared to the analogous zero-order result given by Theorem 3.1, we see that the number of calls to $\mathcal{O}(\cdot)$, namely $T$, is on the same order, and that in both cases the amortized oracle complexity of $\mathcal{O}(\cdot)$ is $O(m)$. The difference between the settings is that the first-order oracle instantiation sets $m$ to be $\widetilde{\Omega}(d^2)$ times smaller than its zero-order counterpart, and hence we gain this multiplicative factor in the overall oracle complexity. It is interesting to compare this gain to non-private optimization, where the ratio between zero- and first-order oracle complexities is $\Theta(d)$ (Duchi et al., 2015; Kornowski & Shamir, 2024), whereas here we obtain an even larger gap in favor of gradient-based optimization.

As in Section 3, we will present the basic properties of this oracle. We will then plug these into Algorithm 2, leading to the main result of this section, Theorem 6.1. Corresponding proofs are deferred to Appendix A.

---

[2]In this section we use full-batch size for simplicity, yet using smaller batches (of arbitrary size) and applying privacy amplification by subsampling, yields the same results up to constants.

**Algorithm 5** First-order instantiation of $\mathcal{O}(z_t)$ in Line 7 of Algorithm 2

1: **Input:** Current iterate $z_t$, time $t \in \mathbb{N}$, period length $\Sigma \in \mathbb{N}$, accuracy parameter $\alpha > 0$, batch sizes $B_1, B_2 \in \mathbb{N}$, gradient validation size $m \in \mathbb{N}$, noise level $\sigma > 0$.
2: **if** $t \bmod \Sigma = 1$ **then**
3:     Sample minibatch $S_t$ of size $B_1$ from unused samples
4:     Sample $y_1, \ldots, y_{B_1} \overset{iid}{\sim} \mathrm{Unif}(\mathbb{B}_\alpha)$
5:     $g_t = \frac{1}{B_1} \sum_{\xi_i \in S_t} \nabla f(z_t + y_i; \xi_i)$
6: **else**
7:     Sample minibatch $S_t$ of size $B_2$ from unused samples
8:     **for** each sample $\xi_i \in S_t$ **do**
9:         Sample $y_1, \ldots, y_{2m} \overset{iid}{\sim} \mathrm{Unif}(\mathbb{B}_\alpha)$
10:         $\tilde{\nabla} f(z_t; \xi_i) = \frac{1}{m} \sum_{j=1}^{m} \nabla f(z_t + y_j; \xi_i)$
11:         $\tilde{\nabla} f(z_{t-1}; \xi_i) = \frac{1}{m} \sum_{j=m+1}^{2m} \nabla f(z_{t-1} + y_j; \xi_i)$
12:     **end for**
13:     $g_t = g_{t-1} + \frac{1}{B_2} \sum_{\xi_i \in S_t} (\tilde{\nabla} f(z_t; \xi_i) - \tilde{\nabla} f(z_{t-1}; \xi_i))$
14: **end if**
15: **Return** $\tilde{g}_t = g_t + \mathrm{TREE}(\sigma, \Sigma)(t \bmod \Sigma)$

**Lemma 6.3** (Sensitivity). *Consider the gradient oracle $\mathcal{O}(\cdot)$ in Algorithm 5 when acting on two neighboring minibatches $S_t$ and $S'_t$, and correspondingly producing $g_t$ and $g'_t$, respectively. If $t \bmod \Sigma = 1$, then*

$$\|g_t - g'_t\| \leq \frac{L}{B_1}.$$

*Otherwise, conditioned on $g_{t-1} = g'_{t-1}$, we have with probability at least $1 - \delta/2$:*

$$\|g_t - g'_t\| \lesssim \frac{L\sqrt{d}D}{\alpha B_2} + \frac{L\sqrt{\log(dB_2/\delta)}}{\sqrt{m}}.$$

With the sensitivity bound given by Lemma 6.3, we easily derive the privacy guarantee of our algorithm from the Tree Mechanism (Proposition 2.5).

**Lemma 6.4** (Privacy). *Running Algorithm 5 with $m = O(\log(dB_2/\delta)\frac{B_2^2 \alpha^2}{D^2 d})$ and $\sigma = O(\frac{L\sqrt{\log(1/\delta)}}{B_1 \varepsilon} + \frac{LD\sqrt{d\log(1/\delta)}}{\alpha B_2 \varepsilon})$ is $(\varepsilon, \delta)$-DP.*

*Proof.* By Lemma 6.3 and our assignment of $m$, we know that with probability at least $1 - \delta/2$, for any $t$, we have

$$\|g_t - g'_t\| \lesssim \frac{L}{B_1} + \frac{L\sqrt{d}D}{\alpha B_2}.$$

Then the privacy guarantee follows from the Tree Mechanism (Proposition 2.5).

$\square$

We next provide the required oracle variance bound.

**Lemma 6.5** (Variance). *In Algorithm 5, for all $t$ it holds that*

$$\mathbb{E} \|\tilde{g}_t - \nabla F_\alpha(z_t)\|^2 \lesssim \frac{L^2}{B_1} + \frac{L^2 d D^2 \Sigma}{\alpha^2 B_2}$$
$$+ \sigma^2 d \log \Sigma + \frac{L^2 \Sigma}{m B_2},$$
$$\mathbb{E} \|\tilde{g}_t\|^2 \lesssim L^2 + \frac{L^2 d D^2 \Sigma}{\alpha^2 B_2}$$
$$+ \sigma^2 d \log \Sigma + \frac{L^2 \Sigma}{m B_2}.$$

Having set up the required bounds, we can prove our main result for the first-order setting.

*Proof of Theorem 6.1.* The privacy guarantee follows directly from Lemma 6.4, by noting that our parameter assignment implies $B_1 T/\Sigma + B_2 T = O(n)$, hence it allows letting $T = \Theta(n)$ while never re-using samples.

As to the sample complexity, note that our parameter assignment ensures that

$$G_1 = O(G_0 + L),$$
$$G_0 = \widetilde{O}\left(\frac{L}{\sqrt{\Sigma}} + \frac{LDd^{1/2}\Sigma^{1/2}}{\alpha} + \frac{Ld^{1/2}}{\Sigma \varepsilon} + \frac{LDd}{\alpha \varepsilon}\right),$$

similarly to Eq. (7) and Eq. (9) in the proof of Theorem 3.1. The rest of the proof is therefore exactly the same as for Theorem 3.1.

$\square$

# 7. Discussion

In this paper, we studied nonsmooth nonconvex optimization, and proposed differentially private algorithms for this task which return Goldstein-stationary points, improving the previously known sample complexity for this task.

Our single-pass algorithm reduces the sample complexity by at least a $\Omega(\sqrt{d})$ factor compared to the previous such result by Zhang et al. (2024). Furthermore, our result has a sublinear dimension-dependent "non-private" term, which was previously claimed impossible. Moreover, we propose a multi-pass algorithm which preforms sample-efficient ERM with sublinear dimension dependence, and show that it further generalizes to the population.

It is interesting to note that our guarantees are in terms of so-called "approximate" $(\varepsilon, \delta)$-DP, whereas Zhang et al. (2024) derive a Rényi-DP guarantee (Mironov, 2017). This is in fact inherent to our techniques, since we condition on a highly probable event in order to substantially decrease the effective sensitivity of our gradient estimators. Further examining this potential gap between approximate- and Rényi-DP for nonsmooth nonconvex optimization is an interesting direction for future research.

Another important problem that remains open is establishing tight lower bounds for DP nonconvex optimization and perhaps further improving the sample complexities obtained in this paper. We note that the current upper and lower bounds do not fully match even in the smooth setting. In Appendix C, we provide evidence that our upper bound can be further improved, by proposing a computationally-*inefficient* algorithm, which converges to a relaxed notion of stationarity, using even fewer samples than the algorithms we presented in this work.

## Acknowledgements

The authors would like to thank the anonymous reviewers for their helpful suggestions, and in particular, for spotting a miscalculation in the previous version of this work. GK is supported by an Azrieli Foundation graduate fellowship.

## Impact Statement

This paper presents work whose goal is to advance the field of Machine Learning. There are many potential societal consequences of our work, none which we feel must be specifically highlighted here.

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

# A. Proofs

## A.1. Proofs from Section 3

*Proof of Lemma 3.4.* Note that for any $y \in \mathrm{Unif}(\mathbb{S}^{d-1}) : \|\frac{d}{2\alpha}(f(z + \alpha y; \xi) - f(z - \alpha y; \xi))y\| \leq Ld$ due to the Lipschitz assumption. Hence, for any $\xi \in S_t$, by a standard sub-Gaussian bound (Theorem D.2) we have

$$\Pr\left[\|\tilde{\nabla} f(z_t; \xi) - \nabla f_\alpha(z_t; \xi)\| \leq \frac{Ld\sqrt{\log(8dB_1/\delta)}}{\sqrt{m}}\right] \geq 1 - \delta/8B_1. \tag{11}$$

If $t \bmod \Sigma = 1$, then

$$
\begin{aligned}
\|g_t - g'_t\| &= \left\| \frac{1}{B_1}\left(\sum_{\xi \in S_t} \tilde{\nabla} f(z_t; \xi) - \sum_{\xi' \in S'_t} \tilde{\nabla} f(z_t; \xi')\right)\right\| \\
&\leq \left\|\frac{1}{B_1}\left(\sum_{\xi \in S_t} \tilde{\nabla} f(z_t; \xi) - \nabla f_\alpha(z_t; \xi)\right)\right\| + \left\|\frac{1}{B_1}\left(\sum_{\xi \in S_t} \nabla f_\alpha(z_t; \xi) - \sum_{\xi' \in S'_t} \nabla f_\alpha(z_t; \xi')\right)\right\| \\
&\quad + \left\|\frac{1}{B_1}\left(\sum_{\xi' \in S'_t} \tilde{\nabla} f(z_t; \xi') - \nabla f_\alpha(z_t; \xi')\right)\right\|.
\end{aligned}
$$

Further note that $\|\frac{1}{B_1}(\sum_{\xi \in S_t} \nabla f_\alpha(z_t; \xi) - \sum_{\xi' \in S'_t} \nabla f_\alpha(z_t; \xi'))\| \leq 2L/B_1$, hence by Equation Eq. (11) and the union bound,

$$\Pr\left[\left\|\frac{1}{B_1}\left(\sum_{\xi \in S_t} \tilde{\nabla} f(z_t; \xi) - \sum_{\xi' \in S'_t} \tilde{\nabla} f(z_t; \xi')\right)\right\| \geq \frac{2L}{B_1} + \frac{Ld\sqrt{\log(8dB_1/\delta)}}{\sqrt{m}}\right] \leq 1 - \delta/8,$$

which proves the claim in the case when $t \bmod \Sigma = 1$. The other case follows from the same argument. $\square$

*Proof of Lemma 3.5.* By Lemma 3.4 and our assignment of $m$, we know that with probability at least $1 - \delta/2$, the sensitivity of all $t$ is bounded by $O(\frac{L}{B_1} + \frac{L\sqrt{d}D}{\alpha B_2})$, namely for all $t$:

$$\|g_t - g'_t\| \lesssim \frac{L}{B_1} + \frac{L\sqrt{d}D}{\alpha B_2}.$$

Then the privacy guarantee follows from the Tree Mechanism (Proposition 2.5). $\square$

*Proof of Lemma 3.6.* First, note that by Proposition 2.5 and the facts that $\mathbb{E}[g_t] = \nabla F_\alpha(z_t)$ and $\|\nabla F_\alpha(z_t)\| \leq L$, we get

$$\mathbb{E}\|\tilde{g}_t\|^2 \lesssim \mathbb{E}\|g_t\|^2 + d\sigma^2 \log \Sigma \lesssim \mathbb{E}\|g_t - \nabla F_\alpha(z_t)\|^2 + L^2 + d\sigma^2 \log \Sigma,$$

and also

$$\mathbb{E}\|\tilde{g}_t - \nabla F_\alpha(z_t)\|^2 \lesssim \mathbb{E}\|\tilde{g}_t - g_t\|^2 + \mathbb{E}\|g_t - \nabla F_\alpha(z_t)\|^2 \lesssim d\sigma^2 \log \Sigma + \mathbb{E}\|g_t - \nabla F_\alpha(z_t)\|^2.$$

Therefore, we see that in order to obtain both claimed bounds, it suffices to bound $\mathbb{E}\|g_t - \nabla F_\alpha(z_t)\|^2$. To that end, denote by $t_0 \leq t$ the largest integer such that $t_0 \bmod \Sigma = 1$, and note that $t - t_0 < \Sigma$. Further denote $\Delta_j := g_j - g_{j-1}$. Then we have

$$
\begin{aligned}
\mathbb{E}\|g_t - \nabla F_\alpha(z_t)\|^2 &= \mathbb{E}\left\|g_{t_0} + \sum_{j=t_0+1}^{t} \Delta_j - \left(\sum_{j=t_0+1}^{t}(\nabla F_\alpha(z_j) - \nabla F_\alpha(z_{j-1})) + \nabla F_\alpha(z_{t_0})\right)\right\|^2 \\
&= \underbrace{\mathbb{E}\|g_{t_0} - \nabla F_\alpha(z_{t_0})\|^2}_{(I)} + \sum_{j=t_0+1}^{t} \underbrace{\mathbb{E}\|\Delta_j - (\nabla F_\alpha(z_j) - \nabla F_\alpha(z_{j-1}))\|^2}_{(II)}, \tag{12}
\end{aligned}
$$

where the last equality is due to the cross terms having zero mean. We further see that

$$(I) \lesssim \mathbb{E}\left\| g_{t_0} - \frac{1}{B_1}\sum_{\xi_i \in S_{t_0}} \nabla f_\alpha(z_{t_0};\xi_i) \right\|^2 + \mathbb{E}\left\| \frac{1}{B_1}\sum_{\xi_i \in S_{t_0}} \nabla f_\alpha(z_{t_0};\xi_i) - \nabla F_\alpha(z_{t_0}) \right\|^2$$

$$\lesssim \frac{L^2 d^2}{B_1 m} + \frac{L^2}{B_1}, \tag{13}$$

as well as

$$(II) = \mathbb{E}\left\| \frac{1}{B_2}\sum_{\xi_i \in S_t} (\tilde{\nabla}f(z_j;\xi_i) - \tilde{\nabla}f(z_{j-1};\xi_i)) - (\nabla F_\alpha(z_j) - \nabla F_\alpha(z_{j-1})) \right\|^2$$

$$= \frac{1}{B_2^2}\sum_{\xi_i \in S_t} \mathbb{E}\left\| (\tilde{\nabla}f(z_j;\xi_i) - \tilde{\nabla}f(z_{j-1};\xi_i)) - (\nabla F_\alpha(z_j) - \nabla F_\alpha(z_{j-1})) \right\|^2$$

$$\lesssim \frac{1}{B_2^2}\sum_{\xi_i \in S_t} \Big( \mathbb{E}\left\| \tilde{\nabla}f(z_j;\xi_i) - \nabla f_\alpha(z_j;\xi_i) \right\|^2 + \mathbb{E}\left\| \tilde{\nabla}f(z_{j-1};\xi_i) - \nabla f_\alpha(z_{j-1};\xi_i) \right\|^2$$

$$+ \mathbb{E}\left\| (\nabla f_\alpha(z_j;\xi_i) - \nabla f_\alpha(z_{j-1};\xi_i)) - (\nabla F_\alpha(z_j) - \nabla F_\alpha(z_{j-1})) \right\|^2 \Big)$$

$$\lesssim \frac{L^2 d^2}{m B_2} + \frac{d L^2 D^2}{\alpha^2 B_2}. \tag{14}$$

Plugging Eq. (13) and Eq. (14) into Eq. (12) and recalling that $t - t_0 < \Sigma$ completes the proof. $\qquad\square$

## A.2. Proofs from Section 4

*Proof of Lemma 4.4.* First, it suffices to prove the first bound, as

$$\mathbb{E}\|\tilde{g}_t\|^2 \lesssim \mathbb{E}\|\tilde{g}_t - \nabla F_\alpha^{\mathcal{D}}(z_t)\|^2 + \mathbb{E}\|\nabla F_\alpha^{\mathcal{D}}(z_t)\|^2 \leq \mathbb{E}\|\tilde{g}_t - \nabla F_\alpha^{\mathcal{D}}(z_t)\|^2 + L^2.$$

To that end, let $t_0 \leq t$ be the largest integer such that $t_0 \bmod \Sigma \equiv 1$, and note that $t - t_0 < \Sigma$. Define $\Delta_j := \frac{1}{n}\sum_{\xi_i \in \mathcal{D}}(\tilde{\nabla}f(z_j;\xi_i) - \tilde{\nabla}f(z_{j-1};\xi_i))$. It holds that

$$\mathbb{E}\|\tilde{g}_t - \nabla F_\alpha^{\mathcal{D}}(z_t)\|^2 \leq \underbrace{\mathbb{E}\|g_{t_0} - \nabla F_\alpha^{\mathcal{D}}(z_{t_0})\|^2}_{(I)} + \sum_{j=t_0}^{t} \underbrace{\mathbb{E}\|\Delta_j - (\nabla F_\alpha^{\mathcal{D}}(z_j) - \nabla F_\alpha^{\mathcal{D}}(z_{j-1}))\|^2}_{(II)} + \sum_{j=t_0}^{t} \underbrace{\mathbb{E}\|\chi_j\|^2}_{(III)}.$$

Similar to the proof of Lemma 3.6, we have that

$$(I) = \mathbb{E}\left\| g_{t_0} - \frac{1}{n}\sum_{\xi_i \in \mathcal{D}} \nabla f_\alpha(z_{t_0};\xi_i) \right\|^2 \lesssim \frac{L^2 d^2}{nm},$$

$$(II) = \frac{1}{n^2}\mathbb{E}\Big\| \sum_{\xi_i \in \mathcal{D}} (\tilde{\nabla}f(z_j;\xi_i) - \tilde{\nabla}f(z_{j-1};\xi_i)) - (\nabla \widehat{F}_\alpha^{\mathcal{D}}(z_j) - \nabla \widehat{F}_\alpha^{\mathcal{D}}(z_{j-1})) \Big\|^2$$

$$\lesssim \frac{1}{n^2}\sum_{\xi_i \in \mathcal{D}} \Big( \mathbb{E}\left\| \tilde{\nabla}f(z_j;\xi_i) - \nabla f_\alpha(z_j;\xi_i) \right\|^2 + \mathbb{E}\left\| \tilde{\nabla}f(z_{j-1};\xi_i) - \nabla f_\alpha(z_{j-1};\xi_i) \right\|^2 \Big)$$

$$\lesssim \frac{L^2 d^2}{mn},$$

$$(III) \leq d\sigma_1^2 + d\sigma_2^2(\Sigma - 1),$$

overall completing the proof.

$\qquad\square$

*Proof of Theorem 4.1.* Setting $m = \frac{L^2 d\Sigma}{n\sigma_1^2} + \frac{L^2 d}{n\sigma_2^2}$, $\sigma_1 = O(\frac{L\sqrt{T\log(1/\delta)/\Sigma}}{n\varepsilon})$ and $\sigma_2 = O(\frac{LD\sqrt{Td\log(1/\delta)}}{\alpha n\varepsilon})$, the privacy guarantee follows from Lemma 4.3. Moreover, by our parameter settings, we have

$$G_0^2 := \mathbb{E}\|\tilde{g}_t - \nabla F_\alpha^{\mathcal{D}}(z_t)\|^2 \lesssim \frac{L^2 dT\log(1/\delta)/\Sigma}{n^2\varepsilon^2} + \frac{L^2 D^2 T d^2\Sigma\log(1/\delta)}{\alpha^2 n^2\varepsilon^2},$$

$$G_1^2 := \mathbb{E}\|\tilde{g}_t\|^2 \lesssim L^2 + \frac{L^2 dT\log(1/\delta)/\Sigma}{n^2\varepsilon^2} + \frac{L^2 D^2 T d^2\Sigma\log(1/\delta)}{\alpha^2 n^2\varepsilon^2}.$$

Therefore, setting $\Sigma = \widetilde{\Theta}(\frac{\alpha}{D\sqrt{d}})$, we see that $G_0 = \widetilde{O}(\frac{L\sqrt{DT}d^{3/4}}{n\varepsilon\sqrt{\alpha}})$ and $G_1 \lesssim L + G_0$. By Proposition 2.6, we also know that

$$\mathbb{E}\|\overline{\partial}_{2\alpha}\widehat{F}^{\mathcal{D}}(x^{\text{out}})\| \le \mathbb{E}\|\overline{\partial}_\alpha\widehat{F}_\alpha^{\mathcal{D}}(x^{\text{out}})\| \le \frac{F_\alpha(x_0) - \inf F_\alpha}{DT} + \frac{3G_1}{2\sqrt{M}} + G_0$$

$$\le \frac{2\Phi}{DT} + \frac{3G_1}{2\sqrt{M}} + G_0.$$

Recalling that $M = \Theta(\alpha/D)$ and setting $D = \widetilde{\Theta}(\frac{\alpha^2\beta^2}{L^2})$, $T = \widetilde{\Theta}(\frac{\Phi L^2}{\alpha^2\beta^3})$, we have

$$\mathbb{E}\|\overline{\partial}_\alpha\widehat{F}_\alpha^{\mathcal{D}}(x^{\text{out}})\| = \widetilde{O}\left(\frac{\Phi}{DT} + \frac{L\sqrt{D}}{\sqrt{\alpha}} + \frac{L\sqrt{DT}d^{3/4}}{n\varepsilon\sqrt{\alpha}}\right)$$

$$= \frac{\beta}{2} + \widetilde{O}\left(\frac{Ld^{3/4}\sqrt{\Phi}}{n\varepsilon\sqrt{\alpha\beta}}\right).$$

The latter is bounded by $\beta$ for $n = \widetilde{\Omega}\left(\frac{L\sqrt{\Phi}d^{3/4}}{\varepsilon\alpha^{1/2}\beta^{3/2}}\right)$, hence completing the proof. $\qquad\square$

## A.3. Proofs from Section 5

*Proof of Proposition 5.1.* Applying a gradient uniform convergence bound for Lipschitz objectives over a bounded domain (Mei et al., 2018, Theorem 1), shows that with probability at least $1 - \zeta$, for any differentiable $x \in \mathcal{X}$:

$$\left\|\nabla\widehat{F}^{\mathcal{D}}(x) - \nabla F(x)\right\| = \widetilde{O}\left(L\sqrt{\frac{d\log(R/\zeta)}{n}}\right). \tag{15}$$

Therefore, given any $x \in \mathcal{X}$, let $y_1, \ldots, y_k \in \mathbb{B}(x, \alpha)$ be points satisfying $\overline{\partial}_\alpha\widehat{F}^{\mathcal{D}}(x) = \sum_{i=1}^k \lambda_i\nabla\widehat{F}^{\mathcal{D}}(y_i)$ for coefficients $(\lambda_i)_{i=1}^k \ge 0, \sum_{i=1}^k \lambda_i = 1$ — note that such points exist by definition of the Goldstein subdifferential. Noting that $\sum_{i=1}^k \lambda_i\nabla F(y_i) \in \partial_\alpha F(x)$, and recalling that $\overline{\partial}_\alpha F(x)$ is the minimal norm element of $\partial_\alpha F(x)$, we get that

$$\left\|\overline{\partial}_\alpha F(x)\right\| \le \left\|\sum_{i=1}^k \lambda_i\nabla F(y_i)\right\| = \left\|\sum_{i=1}^k \lambda_i(\nabla\widehat{F}^{\mathcal{D}}(y_i) + v_i)\right\| = (\star)$$

where $v_i := \nabla F(y_i) - \nabla\widehat{F}^{\mathcal{D}}(y_i)$ satisfy $\|v_i\| = \widetilde{O}\left(L\sqrt{\frac{d\log(R/\zeta)}{n}}\right)$ for all $i \in [k]$ by Eq. (15). Hence

$$(\star) \le \left\|\sum_{i=1}^k \lambda_i\nabla\widehat{F}^{\mathcal{D}}(y_i)\right\| + \left\|\sum_{i=1}^k \lambda_i v_i\right\|$$

$$\le \left\|\overline{\partial}_\alpha\widehat{F}^{\mathcal{D}}(x)\right\| + \sum_{i=1}^k \lambda_i\|v_i\|$$

$$\le \left\|\overline{\partial}_\alpha\widehat{F}^{\mathcal{D}}(x)\right\| + \widetilde{O}\left(L\sqrt{\frac{d\log(R/\zeta)}{n}}\right).$$

$\square$

## A.4. Proofs from Section 6

*Proof of Lemma 6.3.* The case when $t \bmod \Sigma = 1$ trivially follows the Lipschitz assumption. Thus we will consider the more involved case. For any $\xi \in S_t$, by a standard sub-Gaussian bound (Theorem D.2) we have

$$\Pr\left[\|\tilde{\nabla}f(z_t;\xi) - \nabla f_\alpha(z_t;\xi)\| \leq \frac{L\sqrt{\log(8dB_2/\delta)}}{\sqrt{m}}\right] \geq 1 - \delta/8B_2,$$

so by the union bound, we get that with probability at least $1 - \delta/8$, for all $\xi_i \in S_t$ :

$$\|\tilde{\nabla}f(z_t;\xi) - \nabla f_\alpha(z_t;\xi)\| \leq \frac{L\sqrt{\log(8dB_2/\delta)}}{\sqrt{m}}. \tag{16}$$

Hence,

$$
\begin{aligned}
\|g_t - g'_t\| &\leq \left\|\frac{1}{B_2}\sum_{\xi \in S_t}\left((\tilde{\nabla}f(z_t;\xi) - \tilde{\nabla}f(z_{t-1};\xi_i)) - (\nabla f_\alpha(z_t;\xi)) - \nabla f_\alpha(z_{t-1};\xi))\right)\right\| \\
&+ \left\|\frac{1}{B_2}\sum_{\xi \in S_t}\left((\nabla f_\alpha(z_t;\xi) - \nabla f_\alpha(z_{t-1};\xi)) - \sum_{\xi' \in S'_t}(\nabla f_\alpha(z_t;\xi') - \nabla f_\alpha(z_t;\xi'))\right)\right\| \\
&+ \left\|\frac{1}{B_2}\sum_{\xi' \in S'_t}\left((\tilde{\nabla}f(z_t;\xi') - \tilde{\nabla}f(z_{t-1};\xi')) - (\nabla f_\alpha(z_t;\xi') - \nabla f_\alpha(z_{t-1};\xi'))\right)\right\| \\
&\lesssim \frac{L\sqrt{d}D}{\alpha B_2} + \frac{L\sqrt{\log(dB_2/\delta)}}{\sqrt{m}},
\end{aligned}
$$

where the last inequality step is due to the smoothness of $f_\alpha$ (Fact 2.3) combined with the fact that $\|z_t - z_{t-1}\| \leq 2D$, and Eq. (16).

$\square$

*Proof of Lemma 6.5.* Applying by Proposition 2.5, we have

$$\mathbb{E}\|\tilde{g}_t - \nabla F_\alpha(z_t)\|^2 \lesssim \mathbb{E}\|\tilde{g}_t - g_t\|^2 + \mathbb{E}\|g_t - \nabla F_\alpha(z_t)\|^2 \lesssim d\sigma^2\log\Sigma + \mathbb{E}\|g_t - \nabla F_\alpha(z_t)\|^2,$$

and also since $\mathbb{E}[g_t] = \nabla F_\alpha(z_t)$ and $\|\nabla F_\alpha(z_t)\| \leq L$, we have

$$\mathbb{E}\|\tilde{g}_t\|^2 \lesssim \mathbb{E}\|g_t\|^2 + d\sigma^2\log\Sigma \lesssim \mathbb{E}\|g_t - \nabla F_\alpha(z_t)\|^2 + L^2 + d\sigma^2\log\Sigma.$$

We therefore see that both claimed bounds will follow from bounding $\mathbb{E}\|g_t - \nabla F_\alpha(z_t)\|^2$.

To that end, denote by $t_0 \leq t$ the largest integer such that $t_0 \bmod \Sigma = 1$, and note that $t - t_0 < \Sigma$. Further denote $\Delta_j := g_j - g_{j-1}$. Then we have

$$
\begin{aligned}
\mathbb{E}\|g_t - \nabla F_\alpha(z_t)\|^2 &= \mathbb{E}\left\|g_{t_0} + \sum_{j=t_0+1}^{t}\Delta_j - \left(\sum_{j=t_0+1}^{t}(\nabla F_\alpha(z_j) - \nabla F_\alpha(z_{j-1})) + \nabla F_\alpha(z_{t_0})\right)\right\|^2 \\
&= \mathbb{E}\|g_{t_0} - \nabla F_\alpha(z_{t_0})\|^2 + \sum_{j=t_0}^{t}\mathbb{E}\|\Delta_j - (\nabla F_\alpha(z_j) - \nabla F_\alpha(z_{j-1}))\|^2, \\
&\lesssim \frac{L^2}{B_1} + \sum_{j=t_0}^{t}\underbrace{\mathbb{E}\|\Delta_j - (\nabla F_\alpha(z_j) - \nabla F_\alpha(z_{j-1}))\|^2}_{(\star)} \tag{17}
\end{aligned}
$$

where the second equality is due to the cross terms having zero mean. Moreover, we have

$$
\begin{aligned}
(\star) &= \mathbb{E} \left\| \frac{1}{B_2} \sum_{\xi \in S_t} (\tilde{\nabla} f(z_j; \xi) - \tilde{\nabla} f(z_{j-1}; \xi)) - (\nabla F_\alpha(z_j) - \nabla F_\alpha(z_{j-1})) \right\|^2 \\
&= \frac{1}{B_2^2} \sum_{\xi \in S_t} \mathbb{E} \left\| (\tilde{\nabla} f(z_j; \xi) - \tilde{\nabla} f(z_{j-1}; \xi)) - (\nabla F_\alpha(z_j) - \nabla F_\alpha(z_{j-1})) \right\|^2 \\
&\lesssim \frac{1}{B_2^2} \sum_{\xi \in S_t} \Big( \mathbb{E} \left\| \tilde{\nabla} f(z_j; \xi) - \nabla f_\alpha(z_j; \xi) \right\|^2 + \mathbb{E} \left\| \tilde{\nabla} f(z_{j-1}; \xi) - \nabla f_\alpha(z_{j-1}; \xi) \right\|^2 \\
&\qquad\qquad + \mathbb{E} \left\| (\nabla f_\alpha(z_j; \xi) - \nabla f_\alpha(z_{j-1}; \xi)) - (\nabla F_\alpha(z_j) - \nabla F_\alpha(z_{j-1})) \right\| \Big)^2 \\
&\lesssim \frac{L^2}{mB_2} + \frac{dL^2 D^2}{\alpha^2 B_2},
\end{aligned}
$$

which plugged into Eq. (17) completes the proof by recalling that $t - t_0 \leq \Sigma$.

$\square$

## B. Proof of Proposition 2.6 (O2NC)

We start by noting that the update rule for $\Delta_t$ which is given by

$$
\Delta_{t+1} = \min \left\{ 1, \frac{D}{\|\Delta_t - \eta \tilde{g}_t\|} \right\} \cdot (\Delta_t - \eta \tilde{g}_t)
$$

is precisely the online project gradient descent update rule, with respect to linear losses of the form $\ell_t(\cdot) = \langle \tilde{g}_t, \cdot \rangle$, over the ball of radius $D$ around the origin. Accordingly, recalling that $\mathbb{E} \|\tilde{g}_t - \nabla h(z_t)\|^2 \leq G_1^2$, combining the linearity of expectation with the standard regret analysis of online linear optimization (cf. Hazan, 2016) gives the following:

**Lemma B.1.** *By setting* $\eta = \frac{D}{G_1 \sqrt{M}}$, *for any* $u \in \mathbb{R}^d$ *with* $\|u\| \leq D$ *it holds that*

$$
\mathbb{E}_{\tilde{g}_1, \dots, \tilde{g}_M} \left[ \sum_{m=1}^{M} \langle \tilde{g}_m, \Delta_m - u \rangle \right] \leq \tfrac{3}{2} D G_1 \sqrt{M}.
$$

Back to analyzing Algorithm 2, since $x_t = x_{t-1} + \Delta_t$ it holds that

$$
\begin{aligned}
h(x_t) - h(x_{t-1}) &= \int_0^1 \langle \nabla h(x_{t-1} + s\Delta_t), \Delta_t \rangle \, ds \\
&= \mathbb{E}_{s_t \sim \text{Unif}[0,1]} [\langle \nabla h(x_{t-1} + s_t \Delta_t), \Delta_t \rangle] = \mathbb{E}_{s_t} [\langle \nabla h(z_t), \Delta_t \rangle].
\end{aligned}
$$

Note that $\langle \nabla h(z_t), \Delta_t \rangle = \langle \nabla h(z_t), u \rangle + \langle \tilde{g}_t, \Delta_t - u \rangle + \langle \nabla h(z_t) - \tilde{g}_t, \Delta_t - u \rangle$, so by summing over $t \in [T] = [K \times M]$, we get for any fixed sequence $u_1, \dots, u_K \in \mathbb{R}^d$ :

$$
\begin{aligned}
\inf h \leq h(x_T) &\leq h(x_0) + \sum_{t=1}^{T} \mathbb{E} [\langle \nabla h(z_t), \Delta_t \rangle] \\
&= h(x_0) + \sum_{k=1}^{K} \sum_{m=1}^{M} \mathbb{E} \left[ \langle \tilde{g}_{(k-1)M+m}, \Delta_{(k-1)M+m} - u_k \rangle \right] \\
&\quad + \sum_{k=1}^{K} \sum_{m=1}^{M} \mathbb{E} \left[ \langle \nabla h(z_{(k-1)M+m}), u_k \rangle \right] + \sum_{t=1}^{T} \mathbb{E} [\langle \nabla h(z_t) - \tilde{g}_t, \Delta_t - u \rangle] \\
&\leq h(x_0) + \tfrac{3}{2} K D G_1 \sqrt{M} + \sum_{k=1}^{K} \sum_{m=1}^{M} \mathbb{E} \left[ \langle \nabla h(z_{(k-1)M+m}), u_k \rangle \right] + G_0 D T,
\end{aligned}
$$

where the last inequality follows from applying Lemma B.1 to each $M$ consecutive iterates, and combining the bias bound $\mathbb{E}\|\tilde{g}_t - \nabla h(z_t)\| \leq G_0$ with Cauchy-Schwarz.

Letting $u_k := -D \frac{\sum_{m=1}^{M} \nabla h(z_{(k-1)M+m})}{\left\|\sum_{m=1}^{M} \nabla h(z_{(k-1)M+m})\right\|}$, rearranging and dividing by $DT = DKM$, we obtain

$$\frac{1}{K} \sum_{k=1}^{K} \mathbb{E} \left\| \frac{1}{M} \sum_{m=1}^{M} \nabla h(z_{(k-1)M+m}) \right\| \leq \frac{h(x_0) - \inf h}{DT} + \frac{3G_1}{2\sqrt{M}} + G_0. \tag{18}$$

Finally, note that for all $k \in [K], m \in [M] : \left\| z_{(k-1)M+m} - \overline{x}_k \right\| \leq MD \leq \alpha$ since the clipping operation ensures each iterate is at most of distance $D$ to its predecessor, and therefore $\nabla h(z_{(k-1)M+m}) \in \partial_\alpha h(\overline{x}_k)$. Since the set $\partial_\alpha h(\cdot)$ is convex by definition, we further see that

$$\frac{1}{M} \sum_{m=1}^{M} \nabla h(z_{(k-1)M+m}) \in \partial_\alpha h(\overline{x}_k) \,,$$

and hence by Eq. (18) we get

$$\mathbb{E} \left\| \overline{\partial}_\alpha h(x^{\mathrm{out}}) \right\| = \frac{1}{K} \sum_{k=1}^{K} \mathbb{E} \left\| \overline{\partial}_\alpha h(\overline{x}_k) \right\| \leq \frac{h(x_0) - \inf h}{DT} + \frac{3G_1}{2\sqrt{M}} + G_0.$$

## C. Even better sample complexity via optimal smoothing

In this Appendix, our aim is to provide evidence that the sample complexities of NSNC DP optimization obtained in our work are likely improvable, at least with a computationally inefficient method. This approach is inspired by Lowy et al. (2024), which in the context of smooth optimization, showed significant sample complexity gains using algorithms with exponential runtime. As as we will show, a similar phenomenon might hold for nonsmooth optimization. To that end, we propose a slight relaxation of Goldstein-stationarity, and show it can be achieved using less samples via an exponential time algorithm.

### C.1. Relaxation of Goldstein-stationarity

Recall that $x \in \mathbb{R}^d$ is called an $(\alpha, \beta)$-Goldstein stationary point of an objective $F(x) = \mathbb{E}_\xi[f(x; \xi)]$ if there exist $y_1, \ldots, y_k \in \mathbb{B}(x, \alpha)$ and convex coefficients $(\lambda_i)_{i=1}^{k}$ so that $\|\sum_{i \in [k]} \lambda_i \mathbb{E}_\xi[\nabla f(y_i; \xi)]\| \leq \beta$. Arguably, the two most important properties satisfied by this definition are that

(i) If $f(x; \xi)$ are $L$-smooth, any $(\alpha, \beta)$-stationary point is $O(\alpha + \beta)$-stationary.

(ii) If $\left\|\overline{\partial}_\alpha F(x)\right\| \neq 0$, then $F\left(x - \frac{\alpha}{\|\overline{\partial}_\alpha F(x)\|} \overline{\partial}_\alpha F(x)\right) \leq F(x) - \alpha \left\|\overline{\partial}_\alpha F(x)\right\|$.

The first property shows that Goldstein-stationarity reduces to ("classic") stationarity under smoothness. The second, known as Goldstein's descent lemma (Goldstein, 1977), is a generalization of the classic descent lemma for smooth functions.

It is easy to see that Goldstein-stationarity is equivalent to the existence of a distribution $P$ supported over $\mathbb{B}(x, \alpha)$, such that $\|\mathbb{E}_{\xi, y \sim P}[\nabla f(y; \xi)]\| \leq \beta$. We will now define a relaxation of Goldstein-stationarity that is easily verified to satisfy both of the aforementioned properties.

**Definition C.1.** We call a point $x \in \mathbb{R}^d$ an $(\alpha, \beta)$-*component-wise* Goldstein-stationary point of $F(x) = \mathbb{E}_\xi[f(x; \xi)]$ if there exist distributions $P_\xi$ supported over $\mathbb{B}(x, \alpha)$, such that $\|\mathbb{E}_{\xi, y \sim P_\xi}[\nabla f(y; \xi)]\| \leq \beta$.

In other words, the definition above allows the sampled points $y_1, \ldots, y_k$ in the vicinity of $x$ to vary for different components, and as before, the sampled gradient must have small expected norm. We next show that this relaxed stationarity notion allows improving the sample complexity of DP NSNC optimization.

### C.2. Optimal smoothing and faster algorithm

In the previous sections, given an objective $f$, we used the fact that Goldstein-stationary points of the randomized smoothing $f_\alpha$ correspond to Goldstein-stationary point of $f$, and therefore constructed private gradient oracles of $f_\alpha$, which is

$O(\sqrt{d}/\alpha)$-smooth. Consequently, the sensitivity of the gradient oracle had a $\sqrt{d}$ dimension dependence (as seen in Lemma 3.4), thus affecting the overall sample complexity.

Instead of randomized smoothing, we now consider the Lasry-Lions (LL) smoothing (Lasry & Lions, 1986), a method that smooths Lipschitz functions in a dimension independent manner, which we now recall. Given $h : \mathbb{R}^d \to \mathbb{R}$, denote the so-called Moreau envelope

$$M_\lambda(h)(x) := \min_y \left[ h(y) + \frac{1}{2\lambda} \|y - x\|^2 \right],$$

and the Lasry-Lions smoothing:

$$\tilde{h}_{\lambda\mathrm{LL}}(x) := -M_\lambda(-M_{2\lambda}(h))(x) = \max_z \min_y \left[ h(z) + \frac{1}{4\lambda} \|z - y\|^2 - \frac{1}{2\lambda} \|y - x\|^2 \right]. \tag{19}$$

**Fact C.2.** *[Lasry & Lions, 1986; Attouch & Aze, 1993] Suppose $h : \mathbb{R}^d \to \mathbb{R}$ is $L$-Lipschitz. Then: (i) $\tilde{h}_{\lambda\mathrm{LL}}$ is $L$-Lipschitz; (ii) $|\tilde{h}_{\lambda\mathrm{LL}}(x) - h(x)| \leq L\lambda$ for any $x \in \mathbb{R}^d$; (iii) $\arg\min \tilde{h}_{\lambda\mathrm{LL}} = \arg\min h$; (iv) $\tilde{h}_{\lambda\mathrm{LL}}$ is $O(L/\lambda)$-smooth.*

The key difference between LL-smoothing and randomized smoothing is that the smoothness constant of LL-smoothing is dimension independent. By solving the optimization problem in Eq. (19), it is clear that the values, and therefore gradients, of $\tilde{f}_{\lambda\mathrm{LL}}(x; \xi_i)$ can be obtained up to arbitrarily high accuracy. Notably, it was shown by Kornowski & Shamir (2022) that solving this problem requires, in general, an exponential number of oracle calls to the original function.

Nonetheless, computational considerations aside, a priori it is not even clear that the LL smoothing can help finding Goldstein-stationary points of the original function, which was previously shown for randomized smoothing (Lemma 2.4). This is the purpose of the following result, which we prove:

**Lemma C.3.** *If $h$ is $L$-Lipschitz, then any $\beta$-stationary point of $\tilde{h}_{\lambda\mathrm{LL}}$ is a $(3\lambda L, \beta)$-Goldstein stationary point of $h$.*

Given the lemma above, we are able to utilize smooth algorithms for finding stationary points, and convert the guarantee to Goldstein-stationary points of our objective of interest. Specifically, we will invoke the following result.

**Proposition C.4** (Lowy et al., 2024). *Given an ERM objective $\tilde{F}(x) = \frac{1}{n} \sum_{i=1}^n \tilde{f}(x; \xi_i)$ with $L_0$-Lipschitz and $L_1$-smooth components, and an initial point $x_0 \in \mathbb{R}^d$ such that $\mathrm{dist}(x_0, \arg\min \tilde{F}) \leq R$, there's an $(\varepsilon, \delta)$-DP algorithm that returns $\tilde{x}^{\mathrm{out}}$ with $\mathbb{E} \|\nabla \tilde{F}(\tilde{x}^{\mathrm{out}})\| = \widetilde{O}\left( \frac{R^{1/3} L_0^{2/3} L_1^{1/3} d^{2/3}}{n\varepsilon} + \frac{L_0 \sqrt{d}}{n\varepsilon} \right).$*

We remark that we assume for simplicity that $\mathrm{dist}(x_0, \arg\min \widehat{F}^{\mathcal{D}}) = \mathrm{dist}(x_0, \arg\min \tilde{F}) \leq R$, though the analysis extends to that case where $R$ is the initial distance to a point with sufficiently small loss (e.g., if the infimum is not attained). Overall, by setting $\lambda = \alpha/3L$, and combining Fact C.2, Lemma C.3 and Proposition C.4, we get the following:

**Theorem C.5.** *Under Assumption 2.2, suppose $\mathrm{dist}(x_0, \arg\min \widehat{F}^{\mathcal{D}}) \leq R$. Then there is an $(\varepsilon, \delta)$-DP algorithm that outputs $x^{\mathrm{out}}$ satisfying $(\alpha, \beta)$-component-wise Goldstein-stationarity (in expectation) as long as*

$$n = \widetilde{\Omega}\left( \frac{R^{1/3} L^{4/3} d^{2/3}}{\varepsilon \alpha^{1/3} \beta} \right).$$

### C.3. Proofs from Appendix C

*Proof of Lemma C.3.* Suppose $x$ is a $\beta$-stationary point of $\tilde{h}_{\lambda\mathrm{LL}}$. Let $z^* \in \mathbb{R}^d$ be the solution of the maximization problem defining the LL smoothing. By (Attouch & Aze, 1993, Remark 4.3.e), $z^*$ is uniquely defined, and satisfies

$$\nabla \tilde{h}_{\lambda\mathrm{LL}}(x) \in \partial(M_{2\lambda}(h))(z^*). \tag{20}$$

Further denote $\mathcal{Y}^* := \arg\min_y \left[ h(y) + \frac{1}{4\lambda} \|z^* - y\|^2 \right] \subseteq \mathbb{R}^d$. Rearranging the definition of the Moreau envelope by expanding the square, we see that

$$M_{2\lambda}(h)(z^*) = \frac{1}{4\lambda} \|z^*\|^2 - \frac{1}{2\lambda} \max_y \left[ \langle z^*, y \rangle - 2\lambda h(y) - \frac{1}{2} \|y\|^2 \right],$$

from which we get

$$\partial M_{2\lambda} h(z^*) = \frac{1}{2\lambda} z^* - \frac{1}{2\lambda} \text{conv} \{y^* : y^* \in \mathcal{Y}^*\} = \text{conv} \left\{ \frac{1}{2\lambda}(z^* - y^*) : y^* \in \mathcal{Y}^* \right\}. \tag{21}$$

Furthermore, for any $y^* \in \mathcal{Y}^*$, by first-order optimality it holds that

$$0 \in \partial \left[ h(y^*) + \frac{1}{4\lambda} \|y^* - z^*\|^2 \right] \subseteq \partial h(y^*) + \frac{1}{2\lambda}(y^* - z^*),$$

and therefore

$$\frac{1}{2\lambda}(z^* - y^*) \in \partial h(y^*). \tag{22}$$

By combining Eq. (20), Eq. (21) and Eq. (22) we conclude that

$$\nabla \tilde{h}_{\lambda\text{LL}}(x) \in \partial M_{2\lambda} h(z^*) \subseteq \text{conv} \{\partial h(y^*) : y^* \in \mathcal{Y}^*\} \subseteq \partial_r h(x),$$

where the last holds for $r := \max_{y^* \in \mathcal{Y}^*} \|x - y^*\|$. Therefore, recalling that $\|\nabla \tilde{h}_{\lambda\text{LL}}(x)\| \leq \beta$, all that remains is to bound $r$.

To that end, it clearly holds that $r \leq \|x - z^*\| + \max_{y^* \in \mathcal{Y}^*} \|z^* - y^*\|$. Furthermore, by (Attouch & Aze, 1993, Remark 4.3.e) it holds that $z^* - x = \lambda \nabla \tilde{h}_{\lambda\text{LL}}(x)$ which implies $\|x - z^*\| = \lambda\beta$. As to the second summand, by Eq. (21) it holds that $\max_{y^* \in \mathcal{Y}^*} \|z^* - y^*\| \leq 2\lambda \cdot \max_{g \in \partial M_{2\lambda} h(z^*)} \|g\| \leq 2\lambda L$, by the fact that $M_{2\lambda}(h)$ is $L$-Lipschitz. Overall $r \leq \lambda\beta + 2\lambda L$, and as we can assume without loss of generality that $\beta \leq L$ since otherwise the claim is trivially true (note that all points are $L$ stationary), this completes the proof.

$\square$

# D. Concentration lemma for vectors with sub-Gaussian norm

Here we recall a standard concentration bound for vectors with sub-Gaussian norm, which notably applies in particular to bounded random vectors.

**Definition D.1** (Norm-sub-Gaussian). We say a random vector $X \in \mathbb{R}^d$ is $\zeta$-norm-sub-Gaussian for $\zeta > 0$, if $\Pr[\|X - \mathbb{E} X\| \geq t] \leq 2e^{-t^2/2\zeta^2}$ for all $t \geq 0$.

**Theorem D.2** (Hoeffding-type inequality for norm-sub-Gaussian, Jin et al., 2019). *Let $X_1, \cdots, X_k \in \mathbb{R}^d$ be random vectors, and let $\mathcal{F}_i = \sigma(X_1, \cdots, X_i)$ for $i \in [k]$ be the corresponding filtration. Suppose for each $i \in [k]$, $X_i \mid \mathcal{F}_{i-1}$ is zero-mean $\zeta_i$-norm-sub-Gaussian. Then, there exists an absolute constant $c > 0$, such that for any $\gamma > 0$:*

$$\Pr \left[ \left\| \sum_{i \in [k]} X_i \right\| \geq c \sqrt{\log(d/\gamma) \sum_{i \in [k]} \zeta_i^2} \right] \leq \gamma.$$

