# OpenReview forum: "Improved Sample Complexity for Private Nonsmooth Nonconvex Optimization"
_ICML.cc/2025/Conference — ICML 2025 poster_

### Official Review · Reviewer_rnQ5 · 2025-03-05

**Overall Recommendation:** 3

**Summary:**

This paper provides randomized algorithms for nonconvex nonsmooth optimization under the constraint of differential privacy. The sample complexities of zeroth-order algorithms are greatly improved over those of the previous work (Zhang et al. 2024).
In addition, the author further extended the methodology with first-order algorithms to reduce the oracle complexity.

## update after rebuttal
The author's reply makes sense, so I keep my score.

**Claims And Evidence:**

Strength:

* The sample complexity of zeroth-order algorithms greatly improves over existing work (Zhang et al. 2024).

* The generalization from ERM to excess population loss of nonconvex nonsmooth optimization is a novel and interesting result.

Weakness:

* Although the sample complexity of the proposed zeroth-order algorithms is greatly reduced, these methods have high oracle complexities due to the choice of a large $m$. The overall oracle complexity is no better than the previous work.

* There are multiple typos in the proofs, which makes them hard to read.

**Essential References Not Discussed:**

Some essential references to differential privacy are not discussed in the paper. For example, the multi-pass algorithm uses the moment account method for privacy composition, which was introduced in the following paper:

* Abadi, M., Chu, A., Goodfellow, I., McMahan, H. B., Mironov, I., Talwar, K., & Zhang, L. (2016, October). Deep learning with differential privacy. In Proceedings of the 2016 ACM SIGSAC conference on computer and communications security (pp. 308-318).

Their proof was enhanced by the following manuscript:

* Kulkarni, J., Lee, Y. T., & Liu, D. (2021). Private non-smooth empirical risk minimization and stochastic convex optimization in subquadratic steps. arXiv preprint arXiv:2103.15352.

**Experimental Designs Or Analyses:**

I do not think experiments need to be provided for this paper.

**Methods And Evaluation Criteria:**

The proposed methods and evaluation criteria are sound and make sense to me.

**Other Comments Or Suggestions:**

See above.

**Other Strengths And Weaknesses:**

See above.

**Questions For Authors:**

See above.

**Relation To Broader Scientific Literature:**

This paper is a really interesting work for private nonconvex nonsmooth optimization, however, there are too many typos in the proof.

**Theoretical Claims:**

I checked the proof of most theorems in the main text and supplementary materials, there are some typos and errors. Most errors are negligible, and some are really confusing.
Let me list some typos as follows:

* In Line 236 Theorem 3.1, should the nominator value of $m$ be $\log(d B_1 \delta)$?

* In Line 286 Theorem 3.1, I am confused that the denominator of the first term is $B_1$ instead of $\sqrt{B_1}$. Taking square root on both sides of Lemma 3.6 will give you $\sqrt{B_1}$.

* In Line 575, the proof of Lemma 3.4, the right-hand side of $\geq$ in the probability seems to miss one term $2L / B_1$

* In Line 602, the term II should sum from $t_0 + 1$ to $t$?

* In Line 608 Eq. (13), subscripts should be $\xi_i$ instead.

* In Line 656, is (III) miss a factor of $d$ in the second term?

---

> ### Author Rebuttal · Authors · 2025-03-30
>
> We thank the reviewer for their work and are encouraged by their appreciation of our results!
> We will incorporate the suggested changes and references into the final version.
>
> We are extremely grateful for the reviewer's careful review and for finding numerous typos. We note that the only change incurred by these straightforward fixes (specifically, due to the $\sqrt{B_1}$ in the denominator in Line 286) will be the non-private term in Theorem 3.1 becoming $\sqrt{d}$ times larger, that is $\sqrt{d}/\alpha\beta^3$. We emphasize that this does not affect the main claims in our paper: Our obtained complexity is still $\Omega(\sqrt{d})$ smaller than previous results, and still breaks the erroneous lower bound claimed in previous work. Moreover, neither of our improved empirical/generalization results later in the paper are affected.
>
> We conjecture that the previously claimed dimension-independent bound $O(1/\alpha\beta^3)$ might still be achievable. The $\sqrt{d}$ factor appears to stem from the randomized smoothing technique we employed. With improved smoothing approaches, it may be possible to eliminate this dimension dependence. In the final version we will accommodate the current fix accordingly, explore this conjecture further, and plan to acknowledge the reviewer for their help.
>
> We also appreciate the references suggested by the reviewer. We will add proper citations to Abadi et al. (2016) for the moment account method for privacy composition, as well as to Kulkarni et al. (2021) for their enhancements to the approach.
>
> Regarding the oracle complexity concern, the reviewer is correct that our choice of large $m$ increases the oracle complexity. As noted in Remark 3.3, our parameter choice prioritizes minimizing sample complexity, which comes at the cost of increased oracle calls. In the final version, we will clarify this trade-off more explicitly and discuss potential approaches to balance sample and oracle complexity for different application needs.

---

### Official Review · Reviewer_Qpoq · 2025-03-12

**Overall Recommendation:** 4

**Summary:**

This paper studies the problem of non-smooth non-convex (NSNC) optimization problem under the constraint of differential privacy (DP). The authors first proposed a zeroth-order and single-pass NSNC-DP algorithm that achieves sample complexity of $O(\frac{1}{\alpha\beta^3}+\frac{d}{\epsilon\alpha\beta^2}+\frac{d^{3/4}}{\epsilon^{1/2}\alpha\beta^{5/2}})$ to find a $(\alpha,\beta)$-Goldstein stationary point. Notably, this result improves from the existing best rate by a factor of $\sqrt{d}$, and the key is an improved zeroth-order gradient estimator that has an improved sensitivity in high probability. In addition, the authors also extend the result to the ERM problem where they proposed a multi-pass algorithm, and connected the result to the stochastic case by showing a reduction of Goldstein stationary point from ERM loss to generalization loss. Finally, the authors proposed a first-order algorithm, showing it achieves the same sample complexity while reducing oracle complexity by $O(d^2)$.

**Claims And Evidence:**

NA

**Essential References Not Discussed:**

NA

**Experimental Designs Or Analyses:**

NA

**Methods And Evaluation Criteria:**

NA

**Other Comments Or Suggestions:**

NA

**Other Strengths And Weaknesses:**

### Strengths

This paper provides very concrete results for DP-NSNC optimization problems. The main result, a zeroth-order single-pass algorithm equipped with an improved gradient estimator, improves the sample complexity by a factor of $O(\sqrt{d})$ compared to the best know rate in the literature. This is a significant result since the optimal rate is yet unknown, and technical novelty of using a low-sensitivity gradient estimator is interesting by itself. Furthermore, the proposed multi-pass zeroth-order algorithm and the first-order algorithm are the first results in the literature.

### Weaknesses

I think the high probability guarantee of the sensitivity implicitly assumes the gradient estimator has norm-sub-Gaussian noise, which is not clearly justified. This is a slightly stronger assumption than standard literature.

Overall, I think this paper greatly contributes to the understanding of DP-NSNC problem, and I'd recommend acceptance.

**Questions For Authors:**

1. Regarding the non-private term in the single-pass bound, do the authors know if $O(1/\alpha\beta^3)$ is the optimal sample complexity for non-private NSNC optimization? Also, could the authors further elaborate on Rmk 6.2 that the oracle complexity of first-order algorithm is $d^2$ smaller than zeroth-order algorithm while they achieve the same sample complexity? Does that imply the sample complexity of zeroth-order algorithm could possibly be further improved?

2. In terms of oracle complexity, it seems that $m\gg d^{3/2}$. Is it true that the oracle complexity of zeroth-order algorithm is larger than the previous algorithm by Zhang et. al. 2024 (where their sample complexity if $\sqrt{d}$ worse, but oracle complexity is fixed to be $d$ per sample)?

3. The privacy guarantee is based on high probability bound of the sensitivity instead of a worse case (almost surely) bound. Would that break the privacy guarantee? For example, in the tail event where sensitivity is unbounded, I think privacy no longer holds since the divergence is no longer bounded. Is there a known result that fixes the issue, e.g. some theorem like privacy guarantee in high probability generalizes to privacy guarantee almost surely.

**Relation To Broader Scientific Literature:**

This work fits in the subfield of DP optimization for NSNC objectives. The proposed zeroth-order algorithm improves the existing best known rate in the literature. Furthermore, this paper also provides a multi-pass algorithm for ERM problem, and a first-order algorithm, which are both the first results in the literature.

**Theoretical Claims:**

The theoretical claims are valid.

---

> ### Author Rebuttal · Authors · 2025-03-30
>
> We thank the reviewer and are encouraged by the positive review!
> We clarify the raised issues below:
>
> Regarding the "weakness" mentioned, we want to clarify that we did not assume anything stronger than previous results. In our proof, we use concentration inequalities for sub-Gaussian vectors, but these vectors are actually bounded due to the Lipschitz assumption (a standard assumption in the literature, also used by Zhang et al. (2024)).
>
> We address the Questions below:
>
> (1) We are not aware of a sample complexity lower bound in the non-private literature for finding a Goldstein-stationary point. Regarding oracle complexity, it is known to be $\Theta(1/\alpha\beta^3)$ (see Cutkosky et al., ICML 2023). As pointed out by Reviewer rnQ5, there is a small bug in our analysis that leads to a worse non-private term $O(\sqrt{d}/\alpha\beta^3)$  after fixing the error (primarily by resetting the hyperparameters). Despite this, our result still breaks upon the zero-order oracle lower bound of previous work. The dimension-dependence likely stems from the randomized smoothing technique we employed. We conjecture that the previously claimed dimension-independent bound $O(1/\alpha\beta^3)$ could be achieved, possibly through an improved smoothing technique. Therefore, we cannot currently determine the full extent to which our results might be further improved.
>
> (2) Yes, this is true. As discussed in Remark 3.3, our choice of m aims to minimize the sample complexity, and we will rewrite this to be more clear that this is indeed at the cost of larger oracle complexity in the zero-order case.
>
> (3) The privacy guarantee always holds - the tail event is accounted for in the 'delta' in the $(\epsilon,\delta)$-DP definition, which allows the privacy to break with probability at most $\delta$. As the reviewer correctly guessed, this approach is indeed standard in the DP literature, and we also discuss this in the Discussion section.

---

### Official Review · Reviewer_bTS4 · 2025-03-14

**Overall Recommendation:** 4

**Summary:**

This paper presents novel differentially private (DP) optimization algorithms for nonsmooth and nonconvex objectives, with a focus on achieving Goldstein-stationary points while improving sample complexity. The authors introduce a single-pass algorithm that improves the sample complexity by a factor of $\sqrt{d}$ over previous results, as well as a multi-pass algorithm that further reduces sample complexity while preserving privacy guarantees.

**Claims And Evidence:**

The claim presented in this paper appears to be clear and correct.

**Essential References Not Discussed:**

The paper appears to provide sufficient references to related work.

**Experimental Designs Or Analyses:**

Not applicable; the proof appears to be correct.

**Methods And Evaluation Criteria:**

The privacy and utility guarantees of the algorithms have been rigorously proven.

**Other Comments Or Suggestions:**

I did not identify any noticeable typographical errors in the paper.

**Other Strengths And Weaknesses:**

Strength:

* This paper introduces novel algorithms and improves the sample complexity for differentially private nonsmooth and nonconvex (DP NSNC) optimization problems.

* The application of a concentration argument to tighten the sensitivity bound of the gradient estimator is particularly noteworthy and adds technical depth.

Weakness:
* The algorithmic components, such as the use of the tree-based mechanism, have been explored in prior work. However, the analytical approach adopted in this paper appears to be original and contributes new insights.

**Questions For Authors:**

Overall, I find the paper to be well-written and clear, and I have no further questions at this time.

## update after rebuttal
I keep my positive score.

**Relation To Broader Scientific Literature:**

This paper contributes to the literature on differentially private optimization, particularly in the context of nonsmooth and nonconvex objectives.

**Theoretical Claims:**

The proof appears to be correct.

---

> ### Author Rebuttal · Authors · 2025-03-30
>
> We thank the reviewer and are encouraged by the positive review!

---

### Official Review · Reviewer_haDW · 2025-03-17

**Overall Recommendation:** 3

**Summary:**

The paper provided presents advancements in differentially private (DP) optimization algorithms for stochastic and empirical objectives that are neither smooth nor convex. Here is a summary of the results:

1. Zeroth-order Single-pass algorithm.:
- The proposed $(\epsilon, \delta)$-DP algorithm improves the dependence on the number of dimensions over existing algorithms by a factor of $\Omega( \sqrt{d} )$ in the leading term.

2. Zeroth-order Improved multi-pass algorithm:
- To further reduce the sample complexity, a multi-pass polynomial time algorithm is introduced. This algorithm returns the first known algorithm in private ERM with sublinear dimension-dependent sample complexity for non-smooth non-convex objectives.

3. First-order algorithm with reduced oracle complexity: A first-order (i.e., gradient-based) algorithm is also provided, which maintains the sample complexity but reduces the oracle complexity compared to its zero-order counterpart. This makes the gradient-based algorithm significantly more efficient in terms of oracle calls, confirming it as the best-known method in terms of both sample efficiency and oracle efficiency.

**Claims And Evidence:**

See above.

**Essential References Not Discussed:**

No.

**Experimental Designs Or Analyses:**

Not applicable.

**Methods And Evaluation Criteria:**

Not applicable.

**Other Comments Or Suggestions:**

No.

**Other Strengths And Weaknesses:**

Strengths
-----------------------------------

- The paper provides a clear improvement for a natural problem defined recently by Zhang et al. 2024.

- The paper is well-written and it provides a very clear comparison to previous work.

Weaknesses and Comments
-----------------------------------

1. Although this is a clear improvement of Zhang et al. 2024, the problem studied is esoteric and for this reason it seems that is of interest to a small group of people specialized on this topic.

**Questions For Authors:**

See Weaknesses.

**Relation To Broader Scientific Literature:**

Preserving privacy of the data used to train machine learning models is a very important area with broad interest.

**Theoretical Claims:**

I understood the high-level ideas of the proofs but I was not able to check all the details.

---

> ### Author Rebuttal · Authors · 2025-03-30
>
> We thank the reviewer and are encouraged by the positive review!
>
> Designing optimization algorithms for nonsmooth-nonconvex losses is a topic that gained substantial attention recently mainly due to deep learning applications. Indeed, only few works further provide grounded privacy guarantees when optimizing such losses, as this is a relatively new area of research, yet as the reviewer noted, “Preserving privacy of the data used to train machine learning models is a very important area with broad interest.”   Our work aims to bridge this gap and provide foundational algorithms that may be widely applied as privacy-preserving machine learning continues to grow in importance.

---

### Decision · Program_Chairs · 2025-05-01

**Decision:**

Accept (poster)

**Comment:**

This paper develops a new differentially private algorithm targeting the non-smooth non-convex optimization problem. In the single pass/gradient oracle constrained setting, it improves the dimension dependency of prior work. In the general multi-pass setting/sample constrained setting, it provides an improved sample complexity. Reviewers are generally positive about the significance and presentation of the results.